# Ribonucleotide reductase, a novel drug target for gonorrhea

Jana Narasimhan[1]*, Suzanne Letinski[1†], Stephen P Jung[1], Aleksey Gerasyuto[1‡], Jiashi Wang[1‡], Michael Arnold[1], Guangming Chen[1], Jean Hedrick[1], Melissa Dumble[1§], Kanchana Ravichandran[2], Talya Levitz[3], Chang Cui[4], Catherine L Drennan[2,3,5], JoAnne Stubbe[2,3]*, Gary Karp[1], Arthur Branstrom[1]*

[1]PTC Therapeutics, Inc, South Plainfield, United States; [2]Department of Chemistry, Massachusetts Institute of Technology, Cambridge, United States; [3]Department of Biology, Massachusetts Institute of Technology, Cambridge, United States; [4]Department of Chemistry and Chemical Biology, Harvard University, Cambridge, United States; [5]Howard Hughes Medical Institute, Massachusetts Institute of Technology, Cambridge, United States

**Abstract** Antibiotic-resistant *Neisseria gonorrhoeae* (*Ng*) are an emerging public health threat due to increasing numbers of multidrug resistant (MDR) organisms. We identified two novel orally active inhibitors, PTC-847 and PTC-672, that exhibit a narrow spectrum of activity against *Ng* including MDR isolates. By selecting organisms resistant to the novel inhibitors and sequencing their genomes, we identified a new therapeutic target, the class Ia ribonucleotide reductase (RNR). Resistance mutations in *Ng* map to the N-terminal cone domain of the α subunit, which we show here is involved in forming an inhibited $\alpha_4\beta_4$ state in the presence of the β subunit and allosteric effector dATP. Enzyme assays confirm that PTC-847 and PTC-672 inhibit *Ng* RNR and reveal that allosteric effector dATP potentiates the inhibitory effect. Oral administration of PTC-672 reduces *Ng* infection in a mouse model and may have therapeutic potential for treatment of *Ng* that is resistant to current drugs.

**\*For correspondence:**
jnarasimhan@ptcbio.com (JN);
stubbe@mit.edu (JAnneS);
abranstrom@ptcbio.com (AB)

**Present address:** [†]Bristol Myers Squibb, New Brunswick, United States; [‡]Schrödinger, Inc, New York, United States; [§]PMV Pharmaceuticals, Cranbury, United States

## Editor's evaluation

This paper is of interest to biochemists and those focused on development of novel antibiotics. The authors present two small molecules that specifically target the essential ribonucleotide reductase of the causative agent of gonorrhea. Biochemical, biophysical, and biological data support the efficacy of these molecules both in vitro and in mouse models. Overall, this is a comprehensive study providing insights that may guide the development of new therapies for gonorrhea.

## Introduction

Increasing resistance to current therapeutics against many pathogenic bacteria (*Walsh, 2015*; *Gerasyuto et al., 2018*) requires new treatment options with unique mechanism(s) of action. Recently, we reported the discovery of a series of fused indolyl-containing 4-hydroxy-2-pyridones that inhibited bacterial DNA synthesis by targeting mutant DNA topoisomerases (gyrase, topoisomerase IV) and improved in vitro antibacterial activity against a range of fluoroquinolone resistant Gram-negative strains (*Arnold et al., 2017*; *Gerasyuto et al., 2018*).

The interesting activities of a subset of 4-hydoxy-2-pyridones provided the impetus for synthesis of additional chemotypes with this core (*Figure 1*) and their evaluation for effectiveness against additional pathogenic strains including *Ng* and *N. meningitidis* (*Nm*). One of the global challenges of the

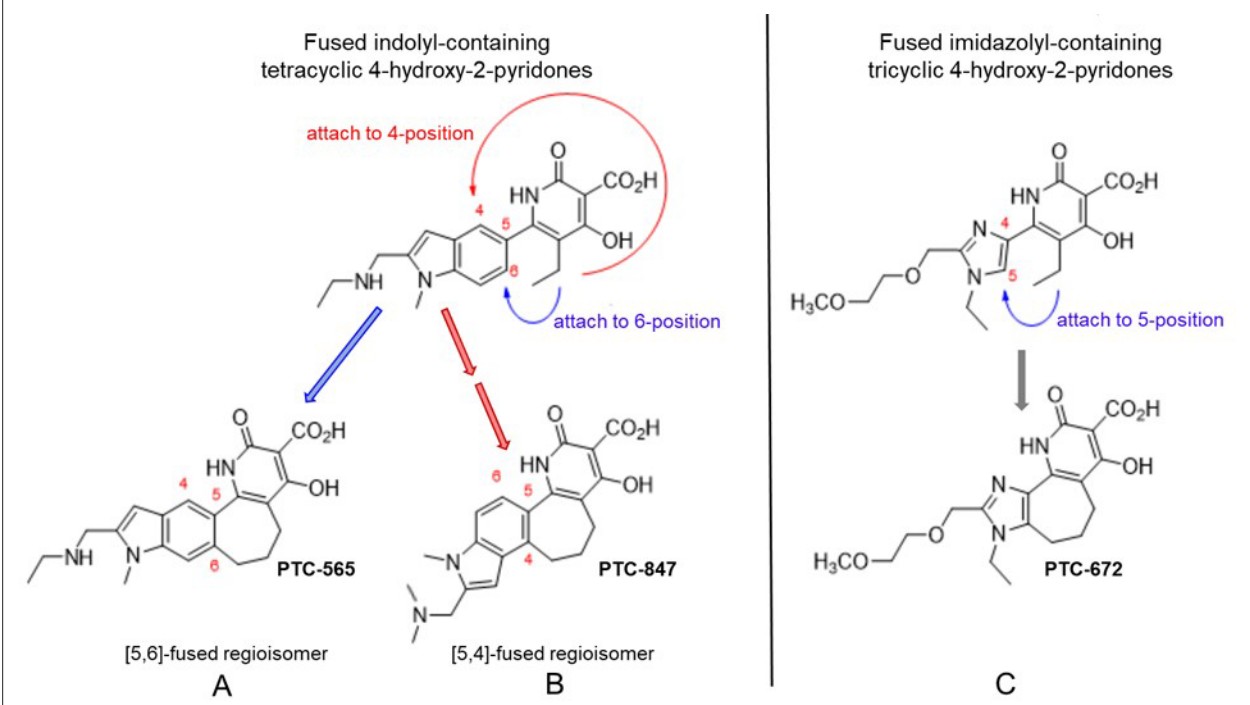

**Figure 1.** Structures of PTC-847 and PTC-672 with 4-hydroxypyridone nucleus. Three methylene units link the 4-hydroxy-2-pyridone core through the indole C-5 and C-4 position yielding two fused seven-membered ring regioisomers with restricted conformations, shown as compounds A and B. The [5,6]-fused regioisomer (PTC-565, **A**) is a broad spectrum DNA gyrase and topoisomerase IV inhibitor targeting Gram-negative pathogens (**Gerasyuto et al., 2018**). The [5,4]-fused regioisomer with a basic dimethylamine group appended to the indole C-2 (PTC-847, **B**) is a potent class Ia RNR inhibitor selective for *Ng*. PTC-672 (**C**), a second selective *Ng* class Ia RNR inhibitor, has an imidazolyl moiety fused to the 4-hydroxy-2-pyridone through a seven-membered ring.

*WHO, 2012* focused on *Ng* isolates. They reported 78 million *Ng* infections worldwide with 90% of infections in low- and middle-income countries. In the US, during 2017–2018,, the rate of reported *Ng* infections increased 5% (583,405 US reported cases in 2018 from *CDC, 2019a*), and the rate increased among both sexes, in all regions of the US, and among all ethnicity groups. *Ng* was reported to be the second most prevalent notifiable sexually transmitted infection in the US (*CDC, 2019a*). The Gonococcal Isolate Surveillance Project (GISP), established by the CDC in 1986, has and continues to monitor trends in antimicrobial susceptibilities of *Ng* strains in the United States (*Kirkcaldy et al., 2016*). Based on reported resistance of *Ng* strains, the CDC currently prescribes a two antibiotic protocol using ceftriaxone (a β-lactam) and azithromycin (a macrolide). Increased reports of resistant isolates have prompted an appeal by the CDC to researchers in the public and private sectors to intensify efforts to develop effective new treatments (*CDC, 2019b*).

We now report that two new 4-hydroxy-2-pyridone pharmacophores (PTC-847 and PTC-672, *Figure 1*) inhibit many strains of *Ng* and MDR *Ng* by targeting biosynthesis of DNA. However, isolation of the *Ng* gyrase and topoisomerase IV reported herein revealed little inhibition by these compounds, in contrast to the related analogs in this class we previously reported (PTC-565, *Figure 1A*). In an effort to identify the target(s) of PTC-847 and PTC-672, we thus decided to select resistant *Ng* organisms to these compounds and to sequence their entire genomes.

Sequencing of several resistant strains revealed single nucleotide changes and amino acid substitutions, all located within the N-terminus of the large subunit (α) of the *Ng* class Ia RNR. RNRs catalyze the de novo conversion of the nucleoside diphosphates adenosine (A), guanosine (G), cytidine (C), and uridine (U) (collectively NDPs) to deoxynucleoside diphosphates (dNDPs) (*Figure 2A*). RNRs are essential in all organisms for DNA biosynthesis because of their role in controlling the relative ratios and amounts of dNTP pools. Imbalance in these pools increases mutation rates, replication and repair errors, and genome instability (*Hofer et al., 2012*; *Aye et al., 2015*; *Greene et al., 2020*).

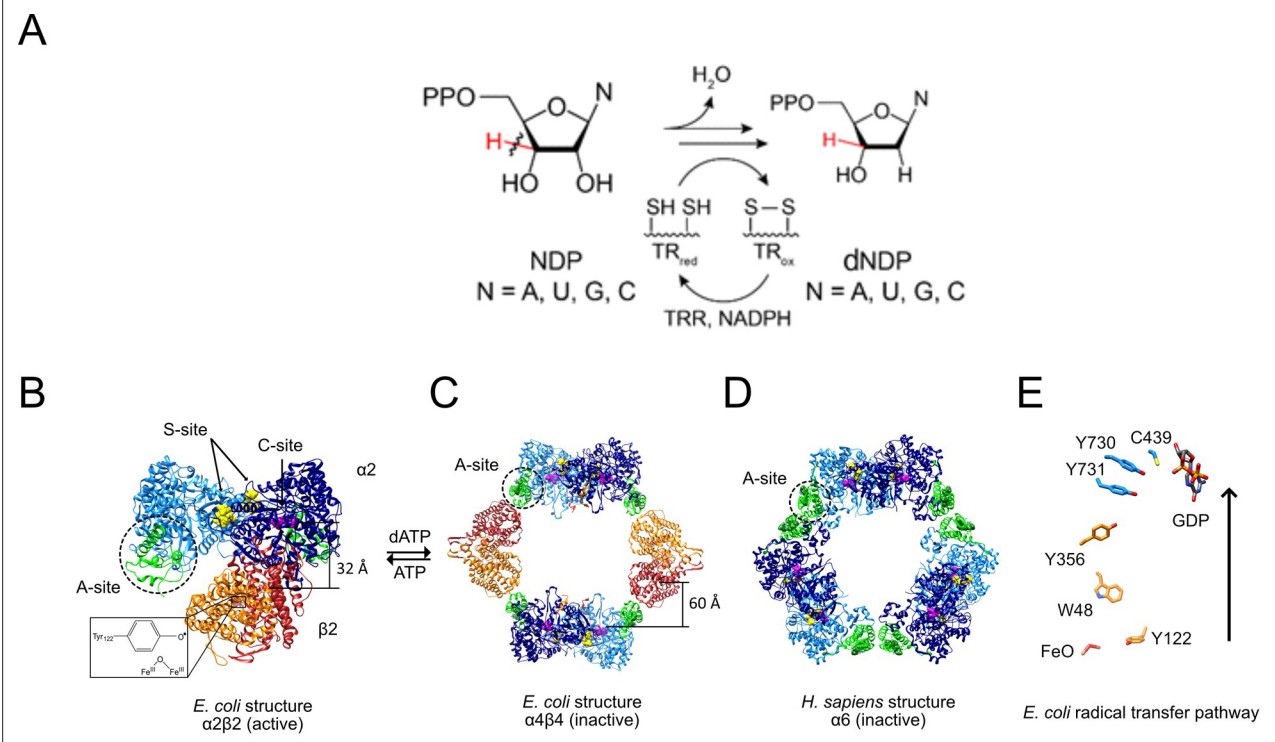

**Figure 2.** RNR reaction (**A**), structures of active and inactive states of Ia RNRs (**B–D**), and the radical transfer pathway (**E**). (**A**) Each of the four NDPs are converted to their corresponding dNDPs, which involves cleavage of the 3'-C-H bond (red) and oxidation of two cysteines to a disulfide. Further turnover requires disulfide reduction by protein reductants thioredoxin (TR) and thioredoxin reductase (TRR). (**B**) A cryo-EM structure of an active *Ec* RNR generated from a double mutant of $\beta_2$ ($F_3Y122\bullet$/E52Q) incubated with $\alpha_2$, substrate GDP (C-site) and allosteric effector TTP (S-site) (**Kang et al., 2020**). (**C**) The inactive dATP-inhibited $\alpha_4\beta_4$ state of *Ec* RNR (**Ando et al., 2011**). (**D**) The inactive dATP inhibited $\alpha_6$ state of the human RNR (**Brignole et al., 2018**). (**E**) The essential diferric-tyrosyl radical cofactor and the components of the radical transfer pathway. A 32 Å distance is shown for the radical transfer pathway in B and 60 Å in C. In B-D, $\alpha_2$ is shown in blue with its N-terminal cone domain in green (A-site) surrounded by a dashed black circle. $\beta_2$ is shown in red and orange.

Most organisms have multiple classes (designated class I, II and III) (**Nordlund and Reichard, 2006**; **Greene et al., 2020**) of RNRs. They all share a common active site located in a structurally homologous protein ($\alpha$, light and dark blue) and a common mechanism of NDP reduction. RNR classification is based on their essential and unique metallo-cofactors in $\beta$ subunit, or one with DOPA radical, a non-metal cofactor (**Greene et al., 2020**) that initiate the complex radical chemistry in $\alpha$ subunit (**Eklund et al., 2001**; **Nordlund and Reichard, 2006**; **Cotruvo and Stubbe, 2011**). *Ng* possesses only one RNR, classified as a class Ia enzyme, that requires a diferric tyrosyl radical cofactor located in its subunit $\beta$ (**Figure 2B** [orange, red], **Eklund et al., 2001**; **Kang et al., 2020**). Class I RNRs use $\alpha$ and $\beta$, two structurally homologous subunits, that are thought to form an active $\alpha_2\beta_2$ complex (**Figure 2B**). The $\alpha$ subunit contains the catalytic site (C-site) and two allosteric effector sites, one involved in the regulation of substrate specificity (S-site) and the other in general enzymatic activity (A-site) (**Eriksson et al., 1997**). The $\beta$ subunit contains the metallo-cofactor required to initiate catalysis upon each NDP reduction event. What is most remarkable about the class Ia RNRs is that the metallo-cofactor in $\beta$ must initiate complex reduction chemistry covering a 32 Å distance ($Y_{122}$ $\beta$ to $C_{439}$ $\alpha$) on each turnover (**Figure 2B and E**).

Some of the Class Ia RNRs, including the human enzyme, can form $\alpha_2\beta_2$ active structures (**Figure 2B**) and are known to be inhibited by dATP (**Nordlund and Reichard, 2006**). The inhibited states, however, are distinct for the *Ec* and human class Ia RNRs (**Figure 2C and D**, respectively). The key to the structures of the inactive states is the N-terminal cone domain of subunit $\alpha$ (green, dotted circles in black). The dATP inhibited *Ec* RNR forms an $\alpha_4\beta_4$ ring structure where the N-terminal cone domain that binds the dATP inhibitor interacts with $\beta_2$ (**Ando et al., 2011**; **Zimanyi et al., 2012**). In the human dATP-inhibited state, the same N-terminal cone domain (green, dotted circles in black) plays a key role in

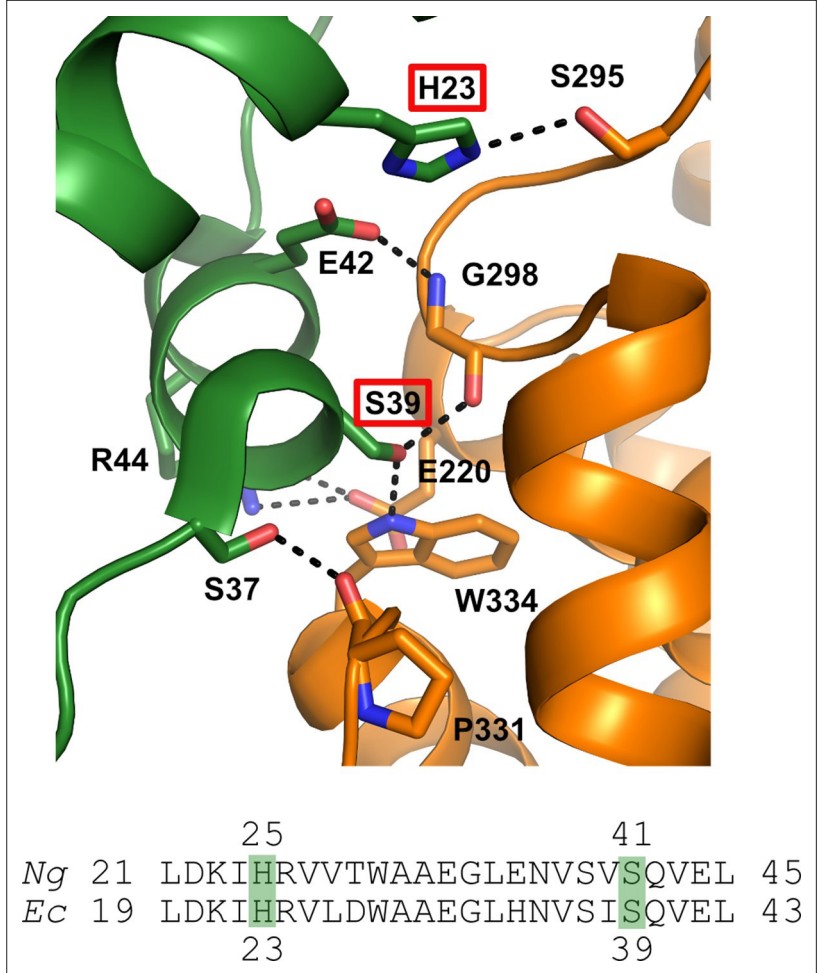

**Figure 3.** Structure of the α / β interface in the inactive a4b4 form of the *E. coli* class Ia RNR where β is in orange and α is in green. Represented are the location of mutations identified in the *Ng* Ia RNR by isolating resistant strains of *Ng* and mapping the mutation sites (H25R and S41L in α). The structure for the *Ec* class Ia RNR is shown with the residues in *Ec* (H23, S39) that correspond to the mutation sites in *Ng* (residues H25 and S41) identified by red boxes (top) (Adapted from *Figure 2*, *Chen et al., 2018*). Also shown is the sequence alignment of α in this region for *Ec* and *Ng*.

formation of a trimer of α dimers in the hexameric $\alpha_6$ ring structure (*Fairman et al., 2011*; *Rofougaran et al., 2006*, *Ando et al., 2011*; *Brignole et al., 2018*; *Greene et al., 2020*). Both inhibited states, by distinct mechanisms, prevent long-range radical initiation between the subunits and hence NDP reduction (*Figure 2E*). The mutations identified in the current studies are located in the N-terminal cone domain.

As introduced above, our present studies reveal that the 4-hydroxy-2-pyridone compounds, PTC-847 and PTC-672 (*Figure 1*), are potent inhibitors of class Ia RNRs, and demonstrate clear antibiotic activity preference toward all *Neisseria* species examined including MDR organisms and the *Nm* M2092 serogroup B (NMSB) reference strain. We also show that the compounds are not active against a panel of ten Gram-negative pathogens and commensal gut microorganisms. Evidence is presented for bactericidal activity against *Ng* through the inhibition of DNA synthesis. This activity is not attributed to inhibition of *Ng* purified DNA gyrase or topoisomerase IV, targets of fluoroquinolone antibiotics. Selection of stable resistant *Ng* isolates for each of these compounds and their subsequent whole genome sequencing led to the identification of mutations in the N-terminal cone domain of the *Ng* Ia RNR, suggesting the potential for inhibition of RNR by these compounds.

Using the structure of the *Ec* $\alpha_4\beta_4$ inhibited state (*Ando et al., 2011*; *Chen et al., 2018*; *Zimanyi et al., 2012*), we show that the resistant mutants map to the N-terminal cone domain of the α subunit

(green), which interacts with the β subunit (orange) of the $\alpha_4\beta_4$ inhibited RNR structure (*Figures 2C and 3*). A high degree of sequence homology between the *Ng* and *Ec* class Ia RNRs suggested that the *Ng* enzyme might also form $\alpha_4\beta_4$ structure, leading us to investigate *Ng* RNR by negative stain EM in this study.

We report purification of the α and β subunits of the *Ng* class Ia RNR, characterization of its activity, and inhibition by dATP and PTC-847 and PTC-672. Using negative stain EM analysis, we demonstrate that *Ng* class Ia RNR can form a dATP-induced $\alpha_4\beta_4$ state and that this state is abrogated by the mutations (H25R and S41L) that cause resistance to these compounds. These studies together suggest that a mode of class Ia RNR inhibition by the 4-hydroxy-2-pyridones might involve binding and potentiation of an inhibited $\alpha_4\beta_4$ state.

To further characterize the potential use of these compounds as a treatment for infections caused by *Ng*, we assessed the efficacy of these compounds in both in vitro and in vivo mouse models. The compounds showed sustained clearance of infection when mice were infected vaginally with either antibiotic-susceptible or the MDR strains of *Ng*. The microbiological and biochemical studies together suggest that we have discovered novel compounds that selectively target the bacterial class Ia RNR for which no clinically useful inhibitors have previously been reported. These compounds also appear to inhibit class Ia RNRs by a new mode of action, which is altering the enzyme's quaternary structure.

## Results

### PTC-847 and PTC-672 are selective inhibitors against *N. gonorrhoeae* in culture

4-Hydroxy-2-pyridone analogs exemplified by PTC-847 and PTC-672 have been synthesized (*Gerasyuto et al., 2016*; *Wang et al., 2016*) to further explore the potential of this pharmacophore as a platform to uncover antibacterial agents targeting *Ng*. The minimum inhibitory concentrations (MIC) of PTC compounds were determined against six *Ng* strains (WHO strains, *Table 1A*) that represent a full range of drug resistant phenotypes relevant to current WHO gonorrhea treatment guidelines (*Unemo et al., 2016*), and ten Gram-negative pathogens and normally occurring gut organisms (*Table 1B*). Two of these strains, WHO F (strain 13477) and WHO K (13479), were selected for a number of subsequent in vitro and in vivo experiments due to their respective sensitivity and resistance to a quinolone antibiotic, ciprofloxacin, that exhibits antibacterial activity by selectively inhibiting DNA synthesis. PTC-847 exhibited potent antibacterial activity against all six *Ng* strains with MICs ranging from 0.05 to 0.1 μg/mL. The weakly immunogenic NMSB strain had a MIC of 0.2 μg/mL (*Table 1A*). PTC-672 exhibited MICs ranging from 0.05 to 0.4 μg/mL against the six *Ng* strains, and a MIC of 0.1 μg/mL against the NMSB strain (*Table 1A*). The compound PTC-565 representing the [5,6] fused ring structure (*Figure 1A*) exhibited antibacterial activity against a broad spectrum of Gram-negative pathogens (*Table 1A and B*); in contrast, PTC-847 and PTC-672 are inactive against the panel of Gram-negative pathogens and normal gut organisms evidenced by MICs ranging from 12.5 to ≥62.5 μg/mL (*Table 1B*).

The unique antibacterial activity against *Ng* strains with PTC-847 and PTC-672 (*Table 1*) provided the impetus to examine them against a panel of 206 *Ng* strains with broad resistance phenotypes collected by the GIST Programme at Public Health England (*Bolt et al., 2016*). Susceptibility testing resulted in $MIC_{90}$ values for PTC-847 and PTC-672 equal to 0.12 μg/mL (determined from values in *Supplementary file 1*). This collection includes strains lacking sensitivity to ceftriaxone and azithromycin, the current standard of care, and underscores that PTC-847 and PTC-672 are highly potent against *Ng*.

At concentrations of 1 X MIC or greater, PTC-847 and PTC-672 exhibit time-dependent cidality against the *Ng* 13477 strain (*Supplementary file 2*). In addition, a time-dependent post antibiotic effect was observed with PTC-847. Suppression of bacterial growth persisted even after the removal of PTC-847 (*Supplementary file 3*).

### PTC-847 targets DNA synthesis, but not DNA topoisomerases

Our previous studies (*Gerasyuto et al., 2018*) established that a variety of [5,6]-fused indolyl-containing 4-hydroxy-2-pyridones (*Figure 1A*) targeted DNA synthesis, more specifically, fluoroquinolone resistant DNA topoisomerases, and inhibited a variety of Gram-negative pathogens. To determine the

**Table 1.** MIC of *Ng* strains (A) and Gram-negative pathogens and gut microorganisms (B).

(A) WHO F does not carry any antimicrobial resistance elements and is considered an antibiotic susceptible strain. WHO G-M strains are resistant to quinolones. WHO K-O strains carry *penA*, *ponA*, *porB1*, and *mtrR* mutations associated with decreased susceptibility to cephalosporins. (*Unemo et al., 2016*). *N. meningitidis* (NMSB) ATCC 13090 is the suggested reference strain for serogroup B (*Clinical and Laboratory Standards Institute, 2004*). (B) This panel was constructed to test for inhibition of Gram-negative pathogens and normally occurring intestinal organisms, where WT = wild type, MDR = multiple drug resistant, and QuinR = quinolone-resistant strains. The *Nm*, Gram-negative, and normal gut organism strains were obtained from the American Type Culture Collection (ATCC), or were kindly provided by MicroMyx, LLC, (MMX), Kalamazoo, MI, or the laboratory of Lynn Zechiedrich (LZ), Baylor College of Medicine, Houston, Texas. PTC-compound susceptibility testing was performed in accordance with the Clinical and Laboratory Standards Institute (CLSI) M07-A9 guideline (*Clinical and Laboratory Standards Institute, 2012*). Genus names: *Acinetobacter* (A), *Klebsiella* (K), *Pseudomonas* (P), *Staphylococcus* (S), *Yersinia* (Y).

Panel A - N. gonorrhoeae and N. meningitidis MIC (µg/mL)

| Strain | | PTC-565 | PTC-847 | PTC-672 |
|---|---|---|---|---|
| **N. gonorrhoeae (WHO F)** | **13477** | **0.1** | **0.05** | **0.05** |
| *N. gonorrhoeae* (WHO G) | 13478 | 0.39 | 0.1 | 0.05 |
| *N. gonorrhoeae* (WHO K) | 13479 | 0.78 | 0.1 | 0.2 |
| *N. gonorrhoeae* (WHO L) | 13480 | 0.19 | 0.05 | 0.2 |
| *N. gonorrhoeae* (WHO M) | 13481 | 0.39 | 0.05 | 0.2 |
| *N. gonorrhoeae* (WHO O) | 13483 | ND | 0.1 | 0.4 |
| *N. meningitidis* (NMSB) | ATCC 13090 | 0.2, 0.39 (two values) | 0.2 | 0.1 |

Panel B - Gram-negative pathogens and normal gut organisms MIC (µg/mL)

| Strain | | PTC-565 | PTC-847 | PTC-672 |
|---|---|---|---|---|
| *A. baumannii* WT | ATCC BAA747 | 0.78 | > 62.5 | ND |
| *A. baumannii* MDR | MMX2240 | 3.1 | 62.5 | > 50 |
| *E. coli* WT | ATCC 25922 | 0.78 | > 62.5 | > 62.5 |
| *E. coli* QuinR | LZ3111 | 0.39 | 31.25 | > 12.5 |
| *K. pneumoniae* WT | ATCC 35657 | 0.78 | 62.5 | ND |
| *K. pneumoniae* MDR | MMX1232 | 15.6 | > 62.5 | ND |
| *P. aeruginosa* WT | ATCC 27853 | 12.5 | > 62.5 | > 62.5 |
| *S. aureus* WT | ATCC 29213 | 12.5 | 62.5 | > 62.5 |
| *S. aureus* MDR | ATCC 700789 | 0.78 | > 12.5 | > 12.5 |
| *Y. pseudotuberculosis* | ATCC 13979 | 1.95 | > 25 | > 25 |

target(s) of PTC-847, we followed the incorporation of radiolabeled precursors into total nucleic acids, DNA, and proteins in the *Ng* 13477 strain. The results revealed that inhibition of DNA but not protein synthesis appears to be responsible for growth inhibition by PTC-847 (*Supplementary file 4*).

We then examined if inhibition of DNA topoisomerases (gyrase and topoisomerase IV) by PTC-847 was responsible for the observed DNA synthesis inhibition. Recombinant *Ng* 13477 gyrase (subunits A and B) and topoisomerase IV (both subunits) were cloned, expressed, and purified and the assay results compared to our previous similar studies on *Ec* DNA gyrase and topoisomerase IV. The PTC-847 half maximal inhibitory concentration (IC$_{50}$) value for *Ng* gyrase was 19 µM and >32 µM for *Ng* topoisomerase IV and both *Ec* topoisomerases (*Table 2*). Furthermore, PTC-847 at 100 µM did not

**Table 2.** PTC-847 does not inhibit *Ng* gyrase or topoisomerase IV in vitro.

Microplate-based supercoiling assays for DNA gyrase and topoisomerase IV were performed as previously described (**Maxwell et al., 2006**). The sequence homology between the *Ng* and *Ec* enzymes are 70% and 75% in gyrase subunit A and B, respectively, and 64% and 65% in topoisomerase IV subunits A and B, respectively. The assays thus followed the *Ec* protocols and are described in detail for the *Ng* enzymes in the Methods. Decatenation assays using purified gyrase AB and topoisomerase IV proteins were performed following the protocols for Profoldin's topoisomerase II and IV DNA decatenation assays (ProFoldin, Hudson, MA) also in Materials and methods.

| | IC50 (µM) | | | | |
| | *E. coli* | | *N. gonorrhoeae* | | Human |
| Compound | Topo IV | Gyrase | Topo IV | Gyrase | Topo II |
| --- | --- | --- | --- | --- | --- |
| Ciprofloxacin | 7 | 0.6 | 18 | 0.4 | – |
| PTC-847 | > 32 | > 32 | > 32 | 19 | > 100 |

appreciably inhibit the activity of human topoisomerase II. As a control, the fluoroquinolone antibiotic ciprofloxacin showed $IC_{50}$ values ranging from 0.4 to 18 µM (**Table 2**). These findings together suggest that topoisomerases are not the primary targets for PTC-847 in *Ng*.

## PTC-847 targets the class Ia RNR large subunit

In an effort to identify the target of inhibition of PTC-847, we used *Ng* 13477 (**Table 1**) to select and isolate a stable PTC-847-resistant isolate (PTC-847[R]) (**Supplementary file 5**). The frequency of resistance for PTC-847[R] at 4 x the MIC was $3.6 \times 10^{-8}$, compared to ciprofloxacin at $1.2 \times 10^{-8}$ and ceftriaxone at $7.7 \times 10^{-9}$ under identical conditions. PTC-847[R] was tested against a wide variety of antibiotics having different modes of action (**Supplementary file 6**). The PTC-847[R] strain was as sensitive to all classes of antibiotics as the wild-type *Ng* 13477 strain, except for susceptibility to PTC-847. The MIC for PTC-847 against *Ng* 13477 was 0.05 µg/mL compared to 15.6 µg/mL for the PTC-847[R] isolate.

To identify the specific PTC-847 molecular target(s), the PTC-847[R] isolate was subjected to whole genome sequencing using the *Ng* 13477 WT strain as the reference. A single C to T transition that results in a serine (S) to leucine (L) amino acid substitution at position 41 (S41L) in the α-subunit of the *Ng* class Ia RNR was identified (**Figure 3**). The gene sequence change was verified by comparing DNA sequences from two of the resistant PTC-847[R] clones against the WT reference strain sequence. The target site conferring resistance to PTC-847 was further confirmed in an isogenic strain (designated PTC-847[S41L]) made by transforming the WT 13477 strain with a DNA fragment containing the α-subunit gene mutation, followed by selection on agar plates containing PTC-847. Hundreds of PTC-847 resistant colonies were obtained when the 13477 strain was transformed with the mutated, but not the WT, α-subunit gene sequence.

A similar strategy using PTC-672 resulted in a stable resistant strain PTC-672[R]. Sequencing the PTC-672[R] class Ia RNR gene revealed a single A to G transition that resulted in a histidine (H) to arginine (R) amino acid substitution at position 25 (H25R) in the α-subunit (**Figure 3**). Thus, both inhibitors target the large subunit of the *Ng* Ia RNR that together with its small subunit are essential for DNA synthesis by catalyzing the conversion of NDPs to dNDPs. In support of the mechanism of inhibition of PTC-847, an increased ratio of NTPs to dNTPs in the susceptible WT 13477 strain was observed upon treatment with the inhibitor at 1 x MIC for 1 hr followed by nucleotide extraction and LC/MS analysis (**Chen et al., 2009**; **Supplementary file 7**).

## Isolation and activity characterization of *Ng* Ia RNR

The active form of the class Ia RNR is $\alpha_2\beta_2$ (**Figure 2B**). The two subunits, however, have weak affinity with the $K_D$ ~0.2 µM for *Ec* and human RNRs; therefore, each subunit was cloned, expressed, and purified independently (**Greene et al., 2020**). Furthermore, assay protocols are optimized for each RNR from each organism. For *Ng* α, an expressed tagged α subunit protein was purified by affinity chromatography and the tag removed to eliminate potential interference with the $\alpha_4\beta_4$ quaternary

**Table 3.** Inhibition of class Ia RNRs in vitro.

Various concentrations of PTC-672 or PTC-847, or 1 mM $N_3$CDP (positive control) were added to 37 °C reaction mixtures containing α, β, ATP, TR, TRR, and NADPH. The mixtures were incubated for either 30 s, 5 min or 15 min prior to initiation of the reaction with 5-[$^3$H]-CDP. Aliquots were quenched at 1, 2, 3 and 4 min in 2% HClO$_4$. All samples were neutralized by the addition of 0.5 M KOH and processed following a standard protocol (*Ravichandran et al., 2020*). Percent inhibition was calculated relative to a 1% DMSO negative control.

| Ia RNR | PTC-672 % Inhibition | PTC-847 % Inhibition |
|---|---|---|
| *Ng* | 78% at 2.5 µM | 93% at 4 µM |
| *Ec* | 92% at 16 µM | 81% at 14 µM |
| Human | 4% at 100 µM | 21% at 100 µM |

structure (*Figure 2B and C*) and enzyme activity. Subunit β was cloned and expressed with an N-terminal (His)$_6$-tag and purified to homogeneity to give 0.7–0.9 diferric tyrosyl radical cofactors (Materials and methods, *Supplementary file 8*). The EPR spectrum of the *Ng* Ia β2 subunit is almost identical to that of the *E. coli* Ia β2 (*Supplementary file 8*). The activity of *Ng* RNR was optimized (*Supplementary file 8*) so that its inhibition by dATP and PTC-847 and PTC-672 could be determined. The activity at 0.1 µM and 1:1(α:β) ratio of subunits was 1300 nmol min$^{-1}$ mg$^{-1}$ with GDP/TTP as substrate and effector, similar to that for *Ec* RNR (2500 nmol min$^{-1}$ mg$^{-1}$ with CDP/ATP as substrate and effector; Materials and methods, *Supplementary file 8*). The apparent $K_m$ is 0.03 µM for *Ng* RNR and as noted above that for *Ec* and human RNRs is ~0.2 µM. The lower $K_m$ and the basis for the drop off in activity at >0.3 µM (*Supplementary file 8*) require further investigation.

All characterized class Ia RNRs have N-terminal cone domains within α-subunit (~100 amino acids) that control enzyme activity (A-site *Figure 2A*, ATP activates and dATP inhibits). Studies of *Ng* RNR activity as a function of dATP concentration (*Supplementary file 9*) reveal a profile very similar to *Ec* RNR, including biphasic kinetics based on dATP binding to both the S-site and A-site (*Hofer et al., 2012*). This result is expected given the 76% sequence identity for α between *Ng* and *Ec*, including the H25 and S41 (*Figure 3*) found altered in *Ng* resistant to the PTC compounds. Human α, on the other hand, shares only 30% sequence identity to *Ng* and does not have residues homologous to

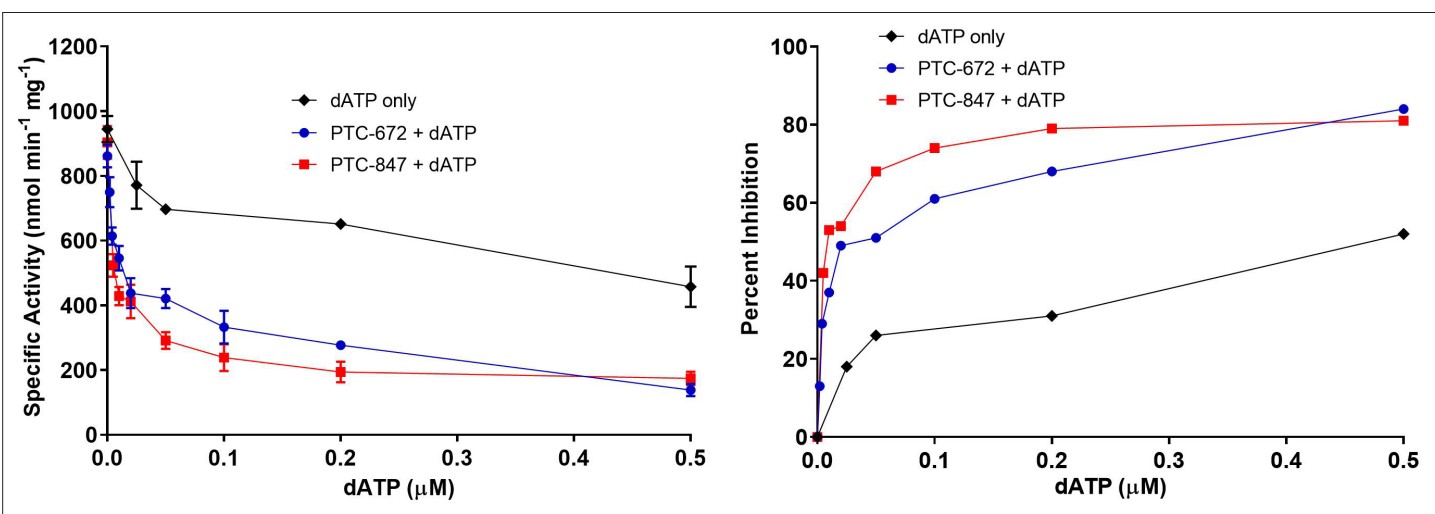

**Figure 4.** Inhibition of RNR by dATP and its potentiation in the presence of PTC-847 and PTC-672. A 1:1 mixture of α and β subunits was incubated with various concentrations of dATP in the presence or absence of 25 nM PTC-672 or 100 nM PTC-847 and assayed for activity spectrophotometrically at 37 °C. The results show that low concentrations of PTC-672 (shown in blue) or PTC-847 (shown in red) potentiate the dATP (shown in black) effect. The activity assay used 0.1 µM α$_2$, 0.1 µM β$_2$, 1 mM GDP, 0.25 mM TTP, 100 µM *Ec* TR, 1 µM *Ec* TRR, 0.2 mM NADPH and various concentrations of dATP (2 nM to 1 µM) with or without 25 nM PTC-647 or 100 nM PTC-847 (n = 2 replicates at each concentration).

The online version of this article includes the following figure supplement(s) for figure 4:

**Figure supplement 1.** Inhibition of *Ng* class Ia RNR by PTC-847 and PTC-672.

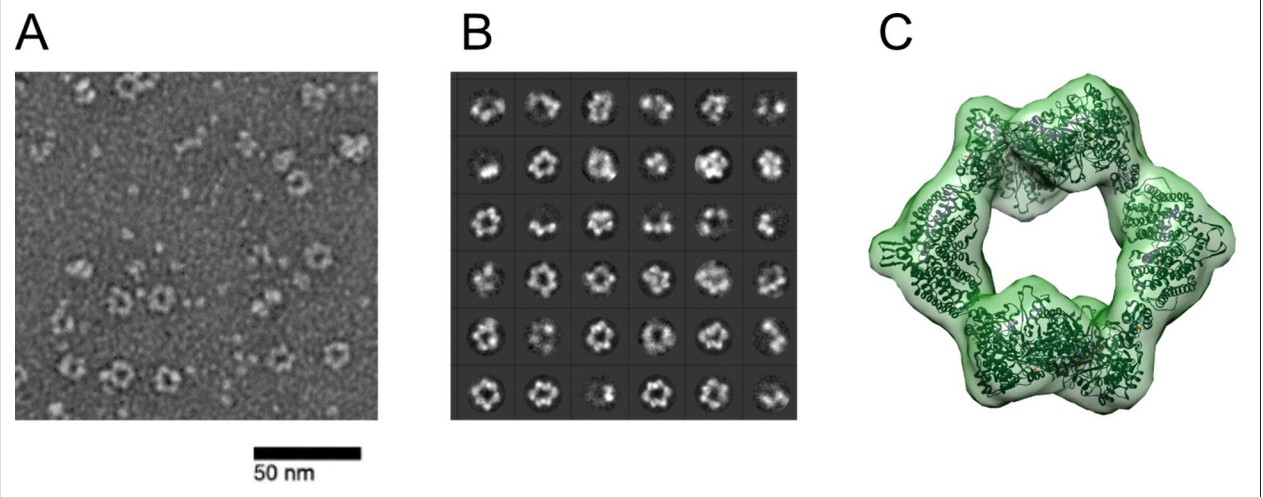

**Figure 5.** *Ng* RNR forms rings under inactivating conditions, as observed with *Ec* RNR. (**A**) Representative negative stain image taken at ×84,000 nominal magnification (1.79 Å/pixel) of 1:1.5 α:β (15 ng/μL final concentration; 0.1 μM $\alpha_2$ and 0.15 μM $\beta_2$) in the presence of 1 mM dATP and 1 mM CDP. Sample was prepared at 4 °C before staining. (**B**) Representative 2D class averages (36 shown of 200 total) from 27,109 picked particles from grids prepared under the same conditions as (**A**). (**C**) Negative stain reconstruction at 21 Å resolution of *Ng* class Ia RNR $\alpha_4\beta_4$ with *Ec* class Ia RNR $\alpha_4\beta_4$ crystal structure fit to the density (PDB 5CNS; *Zimanyi et al., 2016*).Recently (*Levitz et al., 2021*) Levitz et al reported refinement of the Ng a4b4 structure to 4.3 Ang.

either H25 or S41. Inhibition studies with H25R and S41L *Ng*α mutants showed much lower sensitivity to dATP (*Supplementary file 9*).

The *Ng* GDP/TTP (*Supplementary file 8*) and [³H]-CDP/ATP (*Supplementary file 9*) assays were used to assess the ability of PTC-847 and PTC-672 to inhibit *Ng* RNR activity. The results are summarized in *Table 3* and *Figure 4—figure supplement 1*. PTC-847 exhibited 93% inhibition at 4 μM PTC-847, whereas PTC-672 showed 78% inhibition at 2.5 μM. Similar experiments with *Ec* and human RNRs using the [³H]-CDP/ATP assay were also carried out. As expected, based on sequence similarity, the *Ec* RNR is also inhibited, whereas the human enzyme, even at 100 μM inhibitor concentration, still retains substantial activity (*Table 3*).

Inhibition of *Ng* RNR is enhanced in the presence of allosteric inhibitor dATP (*Figure 4* and *Figure 4—figure supplement 1*), suggesting that in vivo, PTC-847 and PTC-672 could act by stabilizing a dATP-induced inhibited state of *Ng* RNR.

## Negative stain EM reveals *Ng* RNR can form $\alpha_4\beta_4$, whereas mutations resulting in PTC-847[R] and PTC-672[R] cannot

We have previously shown using negative stain EM analysis with *Ec* RNR that dATP shifts active RNR into the inhibited $\alpha_4\beta_4$ state (*Figure 2C*), which is readily observed due to the ring structures (*Ando et al., 2011*). Comparable experiments with *Ng* RNR were carried out under very similar conditions to the inhibition studies and the results are shown in *Figure 5*. Panel A shows that, in the presence of 1 mM dATP, ring structures are present. The 2D classifications and 3D reconstruction are shown in *Figure 5B and C*, and reveal that *Ng* Ia RNR, like the *Ec* enzyme, forms the $\alpha_4\beta_4$ ring structure inhibited state induced by dATP.

One hypothesis for RNR inhibition by the 4-hydroxy-2-pyridone derivatives, based on the mapped resistance mutations (*Figures 3 and 5*), is that they potentiate $\alpha_4\beta_4$ formation. To determine if the selected resistant mutants affected the extent of $\alpha_4\beta_4$ formation, PTC-672[H25R] and PTC-847[S41L] α subunits were prepared and purified by the same methods as for the WT proteins and examined as described in *Figure 5*. As shown in *Figure 6*, in the WT control, $\alpha_4\beta_4$ ring structures are clearly visible whereas no such structures are observed with either mutant. An additional control in which ATP (an activator) replaces the dATP inhibitor, also reveals no $\alpha_4\beta_4$ structures for either the WT or mutant proteins (data not shown).

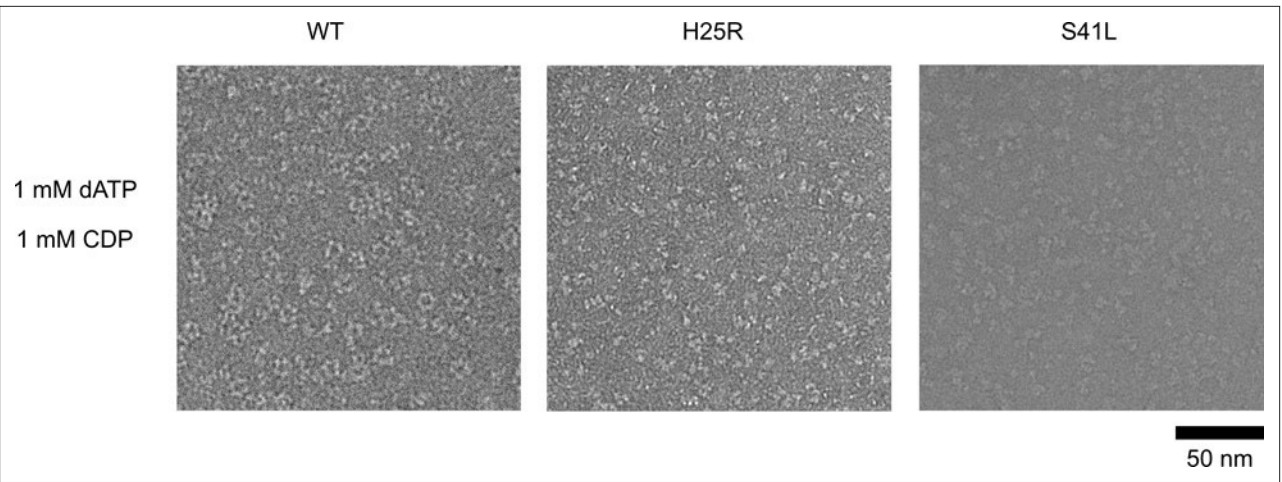

**Figure 6.** *Ng* RNR S41L and H25R variants do not form rings under dATP inhibiting conditions. WT, PTC-672[H25R], or PTC-847[S41L] α subunits were combined with WT β in a 1:1.5 ratio (15 ng/μL final concentration; 0.1 μM α and 0.15 μM β) in the presence of 1 mM dATP and 1 mM CDP and imaged on an F30 (FEI) microscope. Samples were prepared at 4 °C prior to staining.

The online version of this article includes the following figure supplement(s) for figure 6:

**Figure supplement 1.** Inhibition of RNR of the mutants by PTC-672.

**Figure supplement 2.** Representative histograms of mass photometry experiments under differing nucleotide and drug conditions.

**Figure supplement 3.** Percent±standard deviation of $\alpha_4\beta_4$ seen over three replicates of mass photometry experiments under differing nucleotide and drug conditions.

### *Ng* RNR inhibition by PTC-642 and PTC-847 likely involves more than one molecular mechanism

The above data suggest that PTC-847 and PTC-672 may act through the formation/stabilization of *Ng* RNR's $\alpha_4\beta_4$ inactive state, explaining why residue substitutions that prevent $\alpha_4\beta_4$ ring formation would lead to drug resistance in vivo. If ring formation were the only mechanism of inhibition, we would expect that PTC-672 and PTC-847 could not inhibit the H25R and S41L *Ng* RNR variants that cannot form rings. We pursued this idea and found that PTC-672 is far less effective at inhibiting these RNR variants than it is at inhibiting WT *Ng* RNR (*Figure 6—figure supplement 1*). However, a significant amount of inhibition is still occurring (*Figure 6—figure supplement 1*), especially compared to the substantial loss of inhibition for dATP for RNR variants (*Supplementary file 9*) that cannot form rings. This comparison suggests that PTC-672, and likely PTC-847 (not tested), has more than one mode of RNR inhibition.

A mode of inhibition does appear to be related to the formation/stability of $\alpha_4\beta_4$ rings based on the above experiment. Thus, we further pursued whether PTC-672 and PTC-847 act as dATP mimics, that is binding to *Ng* RNR, and leading to increased $\alpha_4\beta_4$ formation. Using mass photometry to evaluate oligomeric states (*Figure 6—figure supplement 2*, *Figure 6—figure supplement 3*), we find an increase in the number of $\alpha_4\beta_4$ rings in the presence of PTC-672/GDP/TTP compared to GDP/TTP alone, but fewer rings than in the dATP control (*Figure 6—figure supplement 2*, *Figure 6—figure supplement 3*). For PTC-847, no increase in rings is observed (*Figure 6—figure supplement 2*, *Figure 6—figure supplement 3*). Based on these results and on the finding that PTC compounds potentiate the inhibitory effects of dATP (*Figure 4*), we postulate that in vivo, PTC-672/PTC-847 are not binding to the dATP site in the cone domain and inducing the conformational changes necessary for $\alpha_4\beta_4$ formation but rather acting synergistically with dATP. Given the differences in structure between PTC-672/PTC-847 and dATP, having different binding sites is not surprising. The exact mechanism(s) by which PTC-672/PTC-847 act in concert with dATP, leading to an additionally inhibited state of *Ng* RNR, awaits further study.

### Assessment of PTC-672 and PTC-847 for treatment of *Ng* infections

In the following sections an in vivo mouse model for vaginal infection with drug susceptible and resistant strains of *Ng* was used to monitor infection clearance. *Ng* 13477 and 13479 (WHO F and

K) susceptible and resistant isolates to PTC-672 (see *Table 1*) were constructed for in vitro and in vivo fitness testing. These isolates were used to monitor the development of resistance to the test compounds in vivo and to assess bacterial survival.

## In vivo studies of *Ng* infections using a mouse model

*Ng* has adapted for the human vaginal tract and will not grow in other species. With administration of high doses of estrogen to ovariectomized female mice and depletion of the normal vaginal bacterial flora, however, mice can be infected with *Ng*. (*Raterman and Jerse, 2019*). To better control the levels of estrogen, these mice are administered estrogen by slow-release pellet two days prior to infection. Concomitantly, they are administered vancomycin, streptomycin, and trimethoprim to deplete commensal flora. These studies are accompanied by administration of streptomycin (Strep) once daily for the remainder of the studies to out-compete the normal flora (for details see Materials and methods).

## *Ng* strain creation for efficacy and fitness studies

As a starting point for susceptibility testing with PTC-672, the *Ng* 13477 strain, which lacks antimicrobial resistance elements (*Unemo et al., 2016*), was tested against the MDR *Ng* 13479 strain that has a high level of resistance to quinolones and carries mutations associated with decreased susceptibility to cephalosporins (*Unemo et al., 2016*). Four new isogenic strains were created for the in vivo studies from *Ng* 13477 and 13479. One set was engineered to have PTC resistance (PTC-672$^{H25R}$) and the second to have resistance to both PTC-672 and streptomycin (Strep$^R$-PTC-672$^{H25R}$). They are designated Iso 13477 Strep$^R$-PTC-672$^{H25R}$ and Iso 13479 Strep$^R$-PTC-672$^{H25R}$. The strains were created using a PCR product containing the H25R α-subunit gene and/or DNA isolated from the streptomycin resistant *Ng* FA1090 laboratory strain.

## In vitro selectivity of PTC-847 and PTC-672

To test if normal intestinal flora are resistant to inhibition by these compounds, PTC-847 and PTC-672 MIC values were determined against a select subset of organisms representing normally occurring intestinal organisms (*Thursby and Juge, 2017*). PTC-847 and PTC-672 spared the commensal bacteria relative to the control antibiotic solithromycin (*Supplementary file 10*), a fourth-generation macrolide with enhanced activity against macrolide-resistant bacteria (*Farrell et al., 2016*). Furthermore, PTC-672's effect on two commensal *Neisseria* strains was also assessed. These isolates were inhibited, but to a lesser extent (approximately 10-fold) than that of *Ng* (*Supplementary file 11*). The reason for the sensitivity differences observed amongst the *Neisseria* species is not known at this time. Based on the target, biological selectivity, and narrow anti-bacterial spectrum of PTC-847 and PTC-672, these compounds may be better tolerated compared with current therapies.

## In vivo efficacy of PTC-672 using the mouse model with susceptible and MDR *Ng* strains

The efficacy of PTC-672 was assessed in the mouse model. On Day 0 of the study, mice were infected vaginally with the *Ng* Iso 13477 Strep$^R$ or Iso 13479 Strep$^R$ strains. Beginning on Day 1, mice were swabbed daily for 1 week to determine the bacterial load. On Day 2, mice were randomized into groups based on Day 1 bacterial load and administered vehicle, ciprofloxacin, ceftriaxone, or PTC-672. Efficacy is defined as complete and sustained clearance of infection at Day 5 post-treatment.

In mice infected with the Iso 13477 Strep$^R$ strain (*Figure 7A*), those that received vehicle had a robust infection for 7 days, with 5/9 mice remaining infected through Day 7. Treatment with a single oral dose of ciprofloxacin at 30 mg/kg resulted in complete clearance of the infection in 10/10 mice (~4 log drop in bacterial infection) within 24 hr following treatment. PTC-672 administered as a single oral dose (10, 15, 20, 25, or 30 mg/kg) resulted in clearance of infection at all doses in 10/10 mice within 24 hr. The clearance of infection by ciprofloxacin or PTC-672 was sustained through Day 7.

In mice infected with the Iso 13479 Strep$^R$ strain (*Figure 7B*), those that received vehicle had a robust infection for 7 days of study, with 8/10 mice remaining infected through Day 7 of study. Treatment with a single intraperitoneal dose of ceftriaxone (100 mg/kg) resulted in complete clearance of the infection in 10/10 mice within 24 hr and clearance of infection was sustained until Day 7. PTC-672 administered as a single oral dose (60 mg/kg) resulted in complete clearance of the infection in

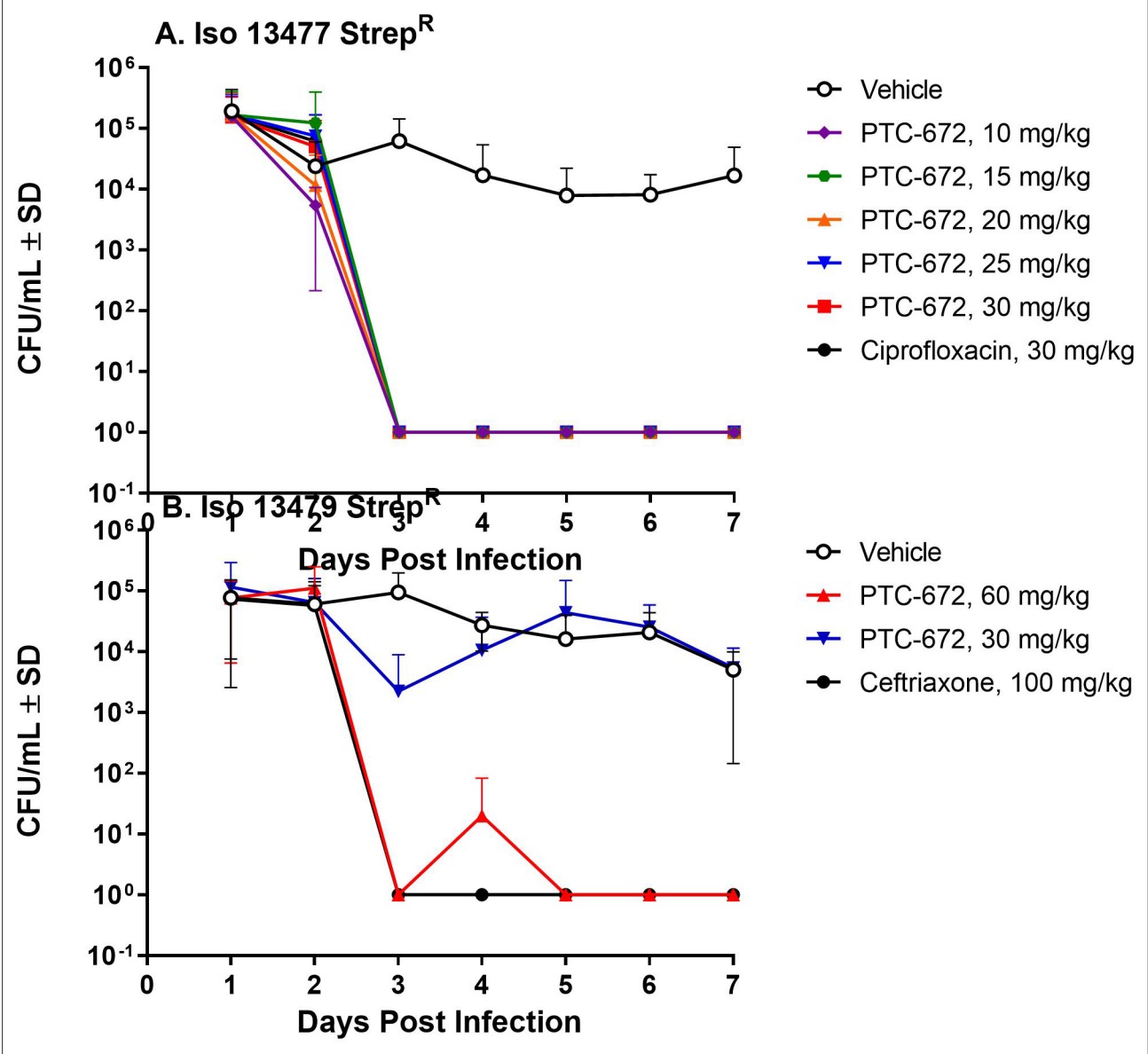

**Figure 7.** PTC-672 in vivo efficacy in a mouse model of gonorrhea. Mice were vaginally inoculated with streptomycin resistant *Ng* strains and swabbed daily over a 1-week period. At each sampling time point, the number of bacteria (CFU/mL) were counted by plating onto selective plates and the mean CFU/mL ± SD values were determined. The mean ± SD values were plotted. (**A**) Symbols: vehicle open black, ciprofloxacin solid black, PTC-672 10 mg/kg purple, 15 mg/kg green, 20 mg/kg orange, 25 mg/kg blue, and 30 mg/kg red. (**B**) Symbols: vehicle open black, ceftriaxone solid black, PTC-672 30 mg/kg blue and 60 mg/kg red.

10/10 mice within 24 hr of dosing and was also sustained until Day 7. At the lower dose of 30 mg/kg PTC-672 resulted in an average 1.46-log drop in bacterial load (8/9 mice cleared infection) 24 hr post-dose. However, infection in several mice re-bounded on subsequent days of the study suggesting that a dose of 30 mg/kg was sub-efficacious in the Iso 13479 Strep[R] model.

### In vitro fitness testing of PTC-672[R]

The relative fitness cost associated with PTC-672 resistance was determined in vitro by the competition strains described above. Competitive fitness assays measure net growth of two populations of isogenic resistant and susceptible strains and account for differences between these populations across the full growth cycle including lag times, exponential growth rates, and stationary phase dynamics.

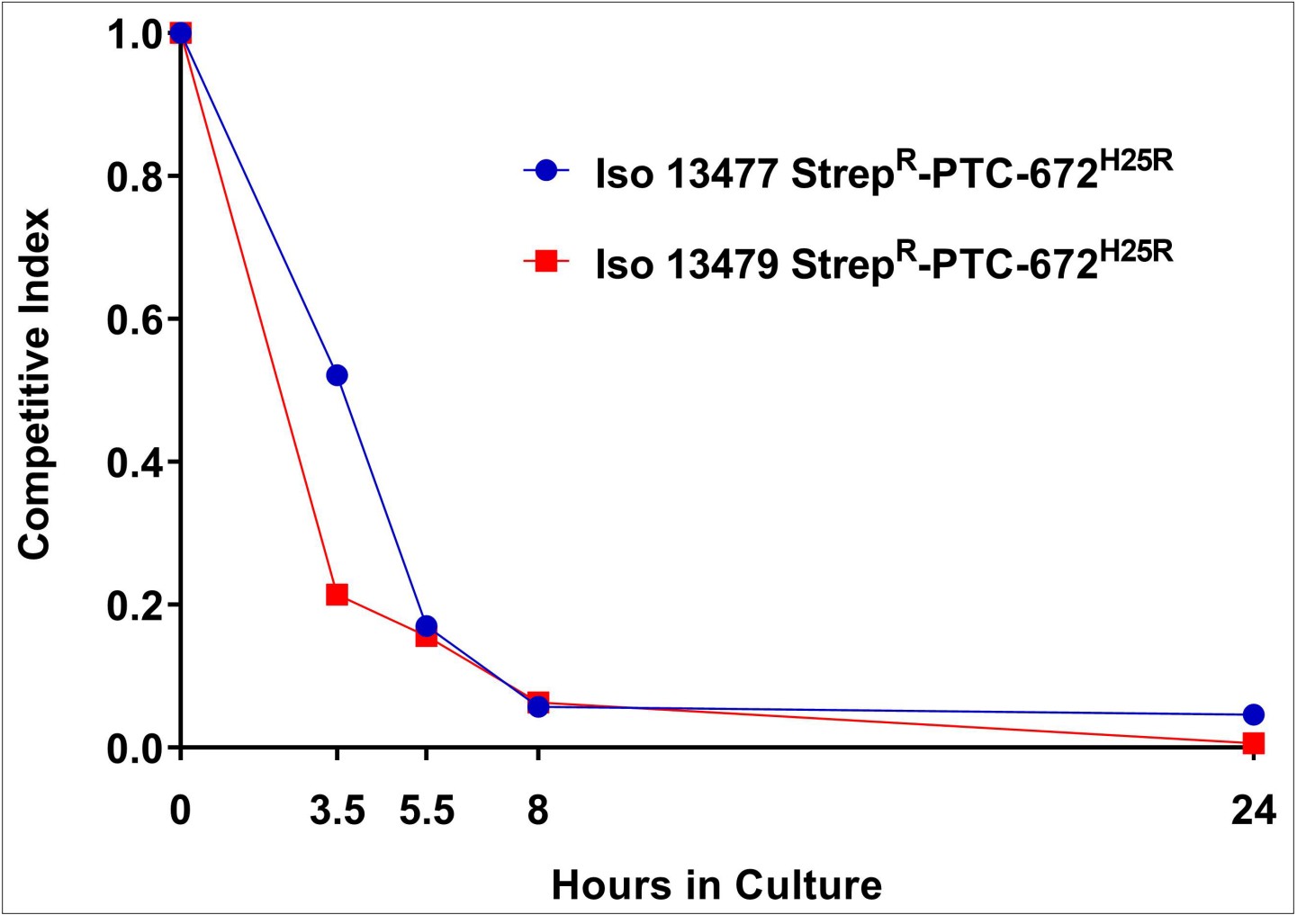

**Figure 8.** H25R mutation in *Ng* RNR reduces fitness in vitro. Iso 13479 Strep^R-PTC-672^H25R (red) and Iso 13477 Strep^R-PTC-672^H25R (blue) were mixed in a 1:1 ratio, inoculated into fresh growth medium, and cultured. At each sampling time point, the numbers of total and resistant bacteria were determined by plating onto non-selective and PTC-672-supplemented plates, respectively. Relative fitness was expressed as the CI (see text).

In this in vitro study, Iso 13477 Strep^R-PTC-672^H25R or Iso 13479 Strep^R-PTC-672^H25R and Iso 13477 Strep^R or Iso 13479 Strep^R were mixed, inoculated into growth medium, and aliquots taken at intervals over a 24 hr incubation period. At each sampling time point, the number of total (susceptible+ resistant) bacteria and resistant bacteria were determined by plating onto non-selective and PTC-672-supplemented plates, respectively.

Relative fitness is expressed as the competitive index (CI), the ratio of bacterial burdens between the PTC-672 resistant and susceptible strains at each time point divided by the baseline ratio at the beginning of the experiment. CI of ~1 indicates that the fitness of the resistant bacteria is similar to that of the parent bacteria, CI >1 indicates that the resistant bacteria are more fit, and CI <1 suggests that the resistant bacteria are less fit in the absence of selection pressure. In this experiment, the CI for *Ng* Iso 13477 Strep^R-PTC-672^H25R and Iso 13479 Strep^R-PTC-672^H25R decreased exponentially to a CI of 0.06 at 8 hr (*Figure 8*), demonstrating that the H25R engineered resistant bacteria are less fit for competitive infection with their respective parent strains while the isolates had similar in vitro doubling times.

### In vivo fitness testing of PTC-672-resistant *Ng*

For these experiments, mice pre-treated as described above were inoculated vaginally with either the isogenic Strep^R-PTC672^H25R strain or a mixture of the Strep^R parent and Strep^R-PTC672^H25R strains and swabbed daily over a 1-week period. At each sampling time point, the relative fitness expressed

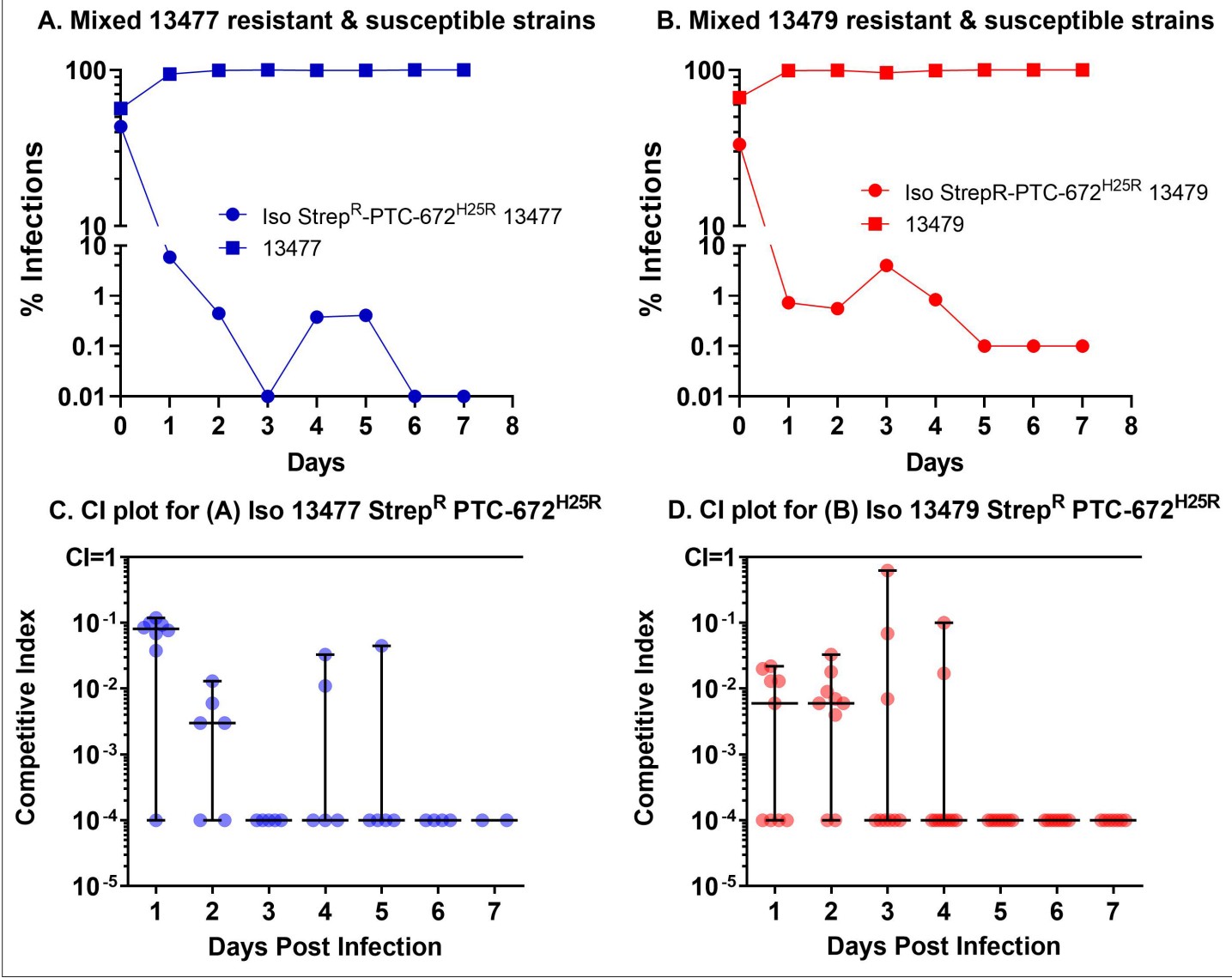

**Figure 9.** H25R mutation in *Ng* α of its Ia RNR reduces fitness in vivo. Mice were vaginally inoculated with the mixed isogenic resistant and susceptible strains and swabbed daily over a one-week period. The CI (plots C and D) was determined at each sampling time point (taken from plots A and B) as described above. The medians ± ranges are shown as solid horizontal lines. (**A**) The H25R mutant strain of 13477 was found to be outcompeted when co-infected with the parent 13477 strain. (**B**) The H25R mutant strain of 13479 (MDR) was found to be outcompeted when co-infected with the parent 13479 strain. (**C**) Competitive index calculations determined from CFUs taken from graph A. (**D**) Competitive index calculations determined from CFUs plotted in graph B.

as the CI was determined by plating onto non-selective and PTC-672-supplemented plates. CI was calculated as the ratio of bacterial burdens between the PTC-672 resistant and susceptible strains at each time point divided by the baseline ratio at the beginning of the experiment.

When infected with only the Iso 13477 Strep$^R$-PTC-672$^{H25R}$ or the Iso 13479 Strep$^R$-PTC-672$^{H25R}$ strain, the infection was similar to that elicited with the Strep$^R$ parent strain. A similar number of mice were infected on Day 1 and the infection level was comparable throughout the study. For fitness testing with mixed strains, the mean and median CIs for *Ng* Iso Strep$^R$-PTC-672$^{H25R}$ 13477 and Iso 13479 Strep$^R$-PTC-672$^{H25R}$ resistant strains decreased to <0.01 by Day 2 post-infection (**Figure 9**).

Thus, although the PTC-672-resistant *Ng* strains infection levels in mice were comparable to those for the isogenic PTC-672 susceptible parent strains, when the two isogenic strains were inoculated in the same anatomical location in vivo, the PTC-672-resistant isolates did not successfully compete with the parent strains. These data suggest that if PTC-672 resistance does arise in *Ng* strains in situ, the

PTC-672 resistant strain would be at a competitive disadvantage and thus less likely to propagate. These results combined with the outcomes of the in vitro fitness study suggest that PTC-672 may lessen the emergence of resistance compared with current therapies.

## Discussion

Novel strategies for developing treatments for highly resistant bacterial pathogens are needed. As a result, there is increased interest in developing single-pathogen therapies against multidrug-resistant organisms. The discovery that the 4-hydroxy-2-pyridone compounds with 5,6 fused ring structures (*Gerasyuto et al., 2016*) exhibited broad-spectrum bactericidal activity against many Gram-negative pathogens by targeting topoisomerases and inhibiting DNA synthesis provided the impetus for further investigation of a general pharmacophore. In conjunction with the reports of increasing MDR in *Ng* by the WHO and CDC, this organism became the focus of our efforts. The discovery that the 4,5-fused pyridines with indolyl and imidazolyl moiety rings (*Figure 1B and C*) were potent and selective inhibitors of *Ng*, *Nm*, and MDR *Ng* (*Table 1*) led to further studies to identify the target(s) of these compounds and the suitability of two compounds (PTC-847 and PTC-672) as potential single organism therapeutics.

We demonstrated that these compounds, although inhibiting DNA synthesis, do not appear to target *Ng* gyrase or topoisomerase IV, but rather uniquely target *Ng* Ia RNR. The dramatic increase in life threatening bacterial infections, coupled to the increase in resistance and MDR to widely used therapeutics (*Walsh, 2015*) makes identification of this new therapeutic target particularly exciting.

This discovery was made through mapping mutants resistant to these compounds to the N-terminal cone domain (100 amino acids) of the *Ec* Ia RNR (*Figure 2B and C*). This domain forms the allosteric activity site responsible for the dATP-induced, inhibited $\alpha_4\beta_4$ state (*Figure 2C*). We have previously shown using negative stain EM analysis with *Ec* Ia RNR that dATP shifts the active $\alpha_2\beta_2$ enzyme into an inhibited $\alpha_4\beta_4$ state, readily observed by its distinct ring structure (*Ando et al., 2011*). Comparable experiments now show that *Ng* Ia RNR enzyme also forms an inhibited ring structure when induced by dATP (*Figure 5*). Studies in vitro (*Chen et al., 2018*) and in vivo (*Ahluwalia et al., 2012*) of *Ec* RNR establish that substitutions of $\alpha/\beta$ interface residues H23 and S39 (H25 and S41 in *Ng* RNR) disrupt this quaternary structure and our preliminary data obtained using EM suggest the same is true for *Ng* RNR (*Figure 5*). The detailed mechanisms by which dATP and PTC compounds inhibit the *Ng* RNR are currently being investigated as many questions about the wild-type and mutant αs and their affinity for $\beta_2$, and the stoichiometry for dATP, PTC-847, and PTC-672 remain unanswered.

Trapping distinct quaternary structures of regulatory proteins with small molecule inhibitors has been successfully exploited with identification of the importance of inhibited protein kinase states in targeting signaling pathways involved in the regulation of cancer cell growth (*Greene et al., 2020*). Recently, biochemical and cell-culture studies have revealed that clorfarabine (ClF), a nucleoside therapeutic used in the treatment of acute lymphoblastic and acute myeloid leukemias, subsequent to its phosphorylation to a di- or trinucleotide (ClFDP, ClFTP) binds to the α subunit of human Ia RNR and potentiates formation of the $\alpha_6$ inhibited state (*Aye and Stubbe, 2011*; *Brignole et al., 2018*). In contrast with the dATP-induced $\alpha_6$ inhibited state (*Fairman et al., 2011*), which rapidly reverts to a mixture of **α m**onomers and dimers on dATP dissociation, both the ClFDP and ClFTP $\alpha_6$ states remain hexameric for some time subsequent to their dissociation (*Brignole et al., 2018*; *Aye and Stubbe, 2011*). Interestingly the *E. coli* Ia RNR, which does not form an $\alpha_6$ state, is not inhibited by ClFTP. While the unique bioinorganic and organic chemistry of RNRs have resulted in targeted inhibition of the active site, $\alpha/\beta$ subunit interactions and formation of the active metallo-cofactor (*Cotruvo and Stubbe, 2011*; *Greene et al., 2020*), the unusual stability of the ClFDP or ClFTP-induced $\alpha_6$ state suggests a novel strategy for selectively targeting this essential enzyme in other organisms.

Despite the structural homology between *Ng* and human α, no appreciable inhibition of the human RNR by PTC-847 or PTC-672 is observed (*Table 3*), consistent with the distinct $\alpha_6$ quaternary structure. The discovery that PTC-847 and PTC-672 inhibited *Ng* but not human Ia RNR suggests that trapping the inhibited quaternary state might be a new approach for therapeutics that selectively target *Ng*.

A number of observations with PTC-847 and –672 further support their potential as new therapeutics. PTC-847 and PTC-672 inhibit all strains of *Ng* and *Nm* (*Table 1*), but examination of a panel of Gram-negative pathogens and a limited subset of normally occurring intestinal flora plus commensal *Ng* strains, were found to be insensitive (*Table 1*, *Supplementary files 10and and 11*). We postulate

that this selectivity may be explained by the observation that *Ng* and *Nm* possess only a single RNR, whereas many microorganisms, including *Escherichia coli, Klebsiella pneumoniae, Bacillus subtilis, Pseudomonas aeruginosa, Bacteroides fragilis, Clostridium difficile, Lactobacillus gasseri, and Fusobacterium nucleatum*, have multiple RNRs (additional class Is [Ib-Ie] or class II and III) (*Lundin et al., 2009*; *Stubbe and Seyedsayamdost, 2018*) that may not be targeted by these compounds (*Supplementary file 12*). Based on the demonstrated target and biological selectivity of PTC-847 and PTC-672, these compounds may be better tolerated compared with current therapies.

We have demonstrated in vivo efficacy of PTC-672 in a mouse model of *N. gonorrhoeae*. When administered as a single oral dose of 60 mg/kg PTC-672, complete clearance of infections was observed with both a susceptible and an MDR strain of *Ng* in 10/10 mice within 24 hr following dosing. We also showed that the H25R–α mutation reduced the competitive advantage of the susceptible and MDR *Ng* strains both in vitro and in vivo mixing experiments. We showed in vivo effectiveness against quinolone resistant and WT *Ng* strains (*Figure 7*) and if resistant isolates arise, they are less fit (*Figures 8 and 9*) and will not likely become dominant. Taken together, these results suggest that treatment with PTC-847, PTC-672 or related analogs exhibiting an acceptable safety profile may be more selective, better tolerated, and lessen the emergence of resistance compared with current therapies.

Resistance of life-threating bacterial infections to most types of antibiotics has been globally emerging over recent years due in part to overprescribing. One possible strategy is to restrict broad-spectrum antibiotic use to treatment of serious MDR infections and to develop novel single-pathogen agents for treating low prevalence infections. However, there is some concern that, given the low annual incidence of infections, single-pathogen agents might not yield a sufficient return on investment to support the development costs. A recent study examined the cost effectiveness of a novel, pathogen-specific agent targeting carbapenem-resistant *A. baumannii* (CRAB) in comparison with the standard-of-care (*Spellberg and Rex, 2013*). The analysis based on annual incidence of sensitive versus resistant infections, cost of treatments, and mortality of infections showed that a single-pathogen agent could provide benefit at a cost well below the traditional benchmarks used to define cost effective therapy. Our compounds fit the profile for a cost-effective single pathogen treatment paradigm for *Ng*.

## Materials and methods
### Antimicrobial susceptibility tests for *Ng* and *Nm*

PTC-565, PTC-847, and PTC-672 susceptibility testing for *Ng* and *Nm* and other Gram-negative pathogens and normal gut organisms was performed in accordance with the CLSI M07-A9 guideline (*Clinical and Laboratory Standards Institute, 2012*). The direct colony suspension procedure was used when testing *Ng* and *Nm*. *Ng* were tested on a gonococcal typing (GC) agar base with 1% defined growth supplement and *Nm* were tested on Mueller-Hinton Agar (MHA) supplemented with 5% sheep blood. Colonies from an overnight chocolate agar culture plate were suspended in 0.9% phosphate buffered saline (PBS), pH 7.0, to a 0.5 McFarland standard. Compounds serially diluted 2-fold were tested in a microtiter assay where all samples tested in duplicate. MIC determinations performed in 96-well plates using FB media (*Takei et al., 2005*). Reaction volumes were 200 µL, with cells in FB media mixed with compound. The final DMSO concentration was limited to 2.5%. Plates were incubated at 35 °C in the presence of 5% $CO_2$ for 48 hr. Turbidity was evaluated by determining the $OD_{600}$ values. The lowest concentration of drug that completely inhibited bacterial growth, in duplicate, was identified as the MIC.

### Bacterial time-dependent kill and post antibiotic effect curves

The time-dependent kill and post antibiotic effect kinetic assays were performed in accordance with the CLSI M26-A guideline (*Clinical and Laboratory Standards Institute, 1999* M26-A 1999). Aliquots from the cultures at the indicated time points were serially diluted 10-fold in PBS ($10^{-1}$ to $10^{-6}$) which were then spotted onto agar plates for 24 hr and the colony forming units per mL (CFU/mL) determined in duplicate. The kill kinetics were represented graphically by plotting the $log_{10}$ CFU/mL against time at each concentration of the compound.

## DNA and protein synthesis assay

The DNA and protein synthesis assays were performed as described (*Jyssum and Jyssum, 1979*). *Ng* 13477 was grown overnight from a single colony in 20 mL fastidious broth (FB) liquid medium at 37 °C with 5% $CO_2$. The culture was then again diluted ~10 fold in FB and grown for 3 h to an OD of 0.4.

For protein assays, 23 mL of this culture was sedimented and washed in 50 mL minimal medium without leucine [10 mM $NaH_2PO_4$ pH 7.0, 12 mM KCl, 6 mM $MgCl_2$, 16 mM $(NH_4)_2SO_4$, 24 mM NaCl, 1 x BD BBL IsoVitaleX Enrichment (Thermo Fisher Scientific, Waltham MA), and 100 µM amino acids minus leucine]. The cells were then resuspended in 22 mL of this media. For the DNA synthesis assay, the cell culture was used directly without refreshing the medium.

Aliquots (200 µL/well) of these cultures were transferred to 96 well microplates. To each well, 5 µL of various 40 x concentrations of PTC-847 in 100% DMSO and a control with 100% DMSO were added (2.5% DMSO final). Then the [$^{14}$C] leucine or [$^{14}$C] uracil (1 µCi/mL), were added and the plates incubated at 35 °C with 5% $CO_2$ atmosphere for 3 h. At the end of the incubation, 90 µL was transferred to the wells of a 96-well filter plate containing an equal volume of 20% tricholoracetic acid (TCA). Another 90 µL was transferred to the wells of a 96 well polypropylene plate containing 10 µL of 3.5 N KOH at 37 °C.

In the former case, protein and total nucleic acids were TCA precipitated at 4 °C for 30 min and the resulting precipitate was collected on filters by centrifugation at 3000 x g for 6 min. In the latter case, RNA was hydrolyzed by KOH treatment at 37 °C overnight and 90 µL of each hydrolysate was transferred to a 96 well filter plate containing an equal volume of 20% TCA (at 4 °C). The remaining DNA was then TCA precipitated at 4 °C for 30 min and collected on filters by centrifugation.

In both workups, the filters were washed with 200 µL ice cold 10% TCA and then dried. The plates were sealed, scintillation cocktail (50 µL/well) was added, and the radioactivity analyzed by scintillation counting (Perkin Elmer).

## *Ng* topoisomerase IV expression and purification

Earlier purification attempts of individual subunits lead to solubility issues of gyrase B, and we therefore chose to co-express the subunits for purification. Plasmids (pET-Duet1) containing the S-tagged *Ng parC* and His-tagged *parE* genes (cloned into sites NcoI/PstI and XhoI/BglII, respectively) were transformed into the recombination-deficient *Ec* (recA) expression strain BLR(DE3) (Novagen) to avoid gene rearrangements. The ParC and ParE proteins were induced and purified using a previously published method (*Pan and Fisher, 1999*).

Single colonies were picked from plates and grown overnight at 37 °C in 50 mL of Luria-Bertani (LB) medium containing the selective antibiotic. A culture (10 mL) of the overnight growth was used to inoculate 450 mL of LB medium containing ampicillin (100 mg/mL) or kanamycin (50 mg/mL). Cells were grown at 30 °C for 3–4 hr, until the optical density at 600 nm reached 0.4–0.6. The culture was transferred to 18 °C and IPTG was added to a final concentration of 0.5 mM, and growth was continued overnight. Bacteria were harvested by centrifugation at 5000 x g for 15 min at 4 °C. The supernatant was discarded, and the bacterial pellet was stored at –80 °C.

The suspension was thawed on ice, and the pellet was resuspended in 20 mL of buffer A (20 mM Tris-HCl [pH 7.9], 500 mM NaCl, 5 mM imidazole, containing EDTA-free complete protease inhibitor (Roche)). Lysozyme (Sigma) and Triton X-100 was added to achieve a final concentration of 0.02% and 0.1% respectively. Incubation was continued on ice for another 30 min and then the mixture was sonicated and centrifuged at 35,000 x g for 60 min.

The supernatant was carefully removed to a 50 mL sterile tube and mixed with 2 mL volume of 50% Ni-NTA resin (Qiagen) which was pre-equilibrated in the buffer A. The tube was shaken gently on a Nutator overnight at 4 °C and then sedimented to remove the Ni-NTA flow through. The Ni-NTA resin was then loaded into a column and washed initially with 20 mL of buffer A, followed by washes with 20 mL of Buffer B (Buffer A plus 50 mM imidazole). The tagged ParC and ParE protein was sequentially eluted with 20 mL buffer C (Buffer A plus 200 mM imidazole) collected as fractions that were stored frozen at –80 °C. The column fractions were examined by western blot. The fractions containing the desired protein were pooled and dialyzed twice for 4 hr against 4 L of 20 mM Tris-HCl (pH 7.9), 250 mM NaCl, 1 mM DTT and 10% glycerol. The dialyzed solution was spun in a microcentrifuge at 16,000 x g for 10 min at 4 °C to remove any precipitate. Purified combined topoisomerase IV was flash frozen in aliquots and stored at –80 °C. The activity of *Ng* topoisomerase IV was measured

using a kDNA decatenation assay and was comparable to commercially available *Ec* topoisomerase IV.

## *Ng* gyrase isolation

*Ng* gyrase isolation was performed using a modification from *Gross et al., 2003*. S-tagged *gyrA* and His-tagged *gyrB* were cloned into NcoI/PstI and XhoI/BglII, sites respectively in pET-Duet1 and transformed into BLR(DE3). Growth and lysozyme treatments were as described above for topoisomerase IV (*Pan and Fisher, 1999*) and the supernatant was mixed with Ni-NTA resin and shaken overnight at 4 °C. The resin was loaded into a column and the flow through containing S-tagged gyrase A was further purified using an S-protein agarose (Sigma) by elution with 20 mM Tris pH 7.5, 300 mM NaCl, 3 M MgCl$_2$.

His-tagged gyrase B was loaded into a Ni-NTA column, washed with 20 mM Tris-HCl [pH 8.0], 300 mM NaCl, 10% glycerol, 0.1% Triton X-100, eluted in the same buffer containing 200 mM imidazole. The purified protein judged by SDS PAGE were pooled and diluted 32-fold in buffer D (50 mM Tris HCl [pH 8.0], 2 mM dithiothreitol (DTT), 1 mM EDTA, 10% glycerol).

Further purification of gyrase A and gyrase B used a mono-Q (GE Healthcare) column in buffer D and a linear gradient elution from 0 to 1 M NaCl. Fractions were analyzed by SDS PAGE and Coomassie blue staining.

The fractions containing the proteins were pooled, concentrated and desalted using Ultrafree-15 Biomax-100 membrane centrifugal filter units (Millipore) in 10 mM Tris pH 7.9, 50 mM KCl, 0.1 mM EDTA, 2 mM DTT. Glycerol was added to the purified protein at a final concentration of 25% and flash frozen in liquid nitrogen, and stored at –80 °C. The protein concentrations were determined by A280 nm and calculated extinction coefficients of *Ng* GyrA (56,400 M$^{-1}$ cm$^{-1}$) and *Ng* GyrB (51,520 M$^{-1}$ cm$^{-1}$). Active gyrase (A$_2$B$_2$) was formed prior to the start of an assay using equal molar amounts (15 µM GyrA and GyrB).

## Bacterial DNA gyrase assays

Microplate-based supercoiling assays for *Ec* or *Ng* DNA gyrase were performed as described (*Maxwell et al., 2006*). *Ec* DNA gyrase was purchased (TopoGEN, Buena Vista, CO). Pierce black streptavidin coated 96-well microplates (Thermo Fisher Scientific, Waltham, MA) were rehydrated and washed 3 x in wash buffer (20 mM Tris-HCl, pH 8.0, 137 mM NaCl, 0.01% bovine serum albumin, 0.05% Tween-20). A total of 100 µL of 500 nM biotinylated triplex forming oligonucleotide (TFO1) was immobilized onto the streptavidin plate. Excess oligonucleotide was washed off using the wash buffer. Enzyme reactions (30 µL) containing 1 µg relaxed plasmid pNO1 DNA (Inspiralis Limited, Norwich, UK) in 35 mM Tris-HCl (pH 7.5), 24 mM KCl, 4 mM MgCl$_2$, 2 mM DTT, 1.8 mM spermidine, 1 mM ATP, 6.5% glycerol, 0.1 mg/ml albumin, and 1 U of *Ec* DNA gyrase (TopoGEN, Buena Vista, CO) or DNA gyrase purified as described above from the *Ng* 13477 strain was incubated at 37 °C for 30 min. A total of 100 µL of TF buffer (50 mM sodium acetate, pH 5.5, 50 mM NaCl, 50 mM MgCl$_2$•6 H$_2$O) was then added to the reaction and the entire mixture transferred to the microplate wells after the TFO1 immobilization process. The microplate was incubated at room temperature for 30 min to allow triplex formation. Unbound plasmid was washed off with 3 × 200 µL of TF buffer, and 200 µL of 1 X SYBR Gold (Invitrogen, Carlsbad, CA) in 10 mM Tris-HCl (pH 8.0), 1 mM EDTA was added and allowed to stain for 20 min. Fluorescence was read using the EnVision Multimode Plate Reader (Perkin-Elmer, Waltham, MA).

## Human DNA topoisomerase II and bacterial DNA topoisomerase IV decatenation assays

These topoisomerases can convert the large network of concatenated kinetoplast DNA (kDNA) from *C. fasciculata* into decatenated DNA. Topoisomerase decatenation assays are based on the principle that decatenated DNA can be separated from concatenated DNA by microfiltration and quantified by fluorescence. Decatenation assays using purified human DNA topoisomerase II (ProFoldin, Hudson, MA), *Ec* DNA topoisomerase IV (ProFoldin, Hudson, MA), and *Ng* DNA topoisomerase IV purified in-house from the *Ng* 13477 strain as described were performed according to ProFoldin's protocols for microplate-based topoisomerase DNA decatenation assays using kDNA (ProFoldin, Hudson, MA) and in-house reagents. The separated decatenated product was stained using the Quant-iT PicoGreen

dsDNA Assay Kit (Invitrogen, Carlsbad, CA) followed by fluorescence quantitation on an EnVision Multimode Plate Reader.

## Nucleotide pools measured in *Ng*

*Ng* 13477 and PTC-847[R] were grown overnight in FB medium and then subcultured. Two *Ng* 13477 and one PTC-847[R] subcultures were allowed to grow for 4 hr. Then PTC-847 (0.05 µg/ml) was added to one of the *Ng* 13477 subcultures and incubated an additional 1 hr at 37 °C with $CO_2$. The bacteria were collected by centrifugation for 15 min at 4000 x g at 4 °C. The pellets were resuspended in 5 mL cold acidified ACN (65% acetonitrile, 35% water, 100 mM formic acid) and extracted on ice for 30 min with periodic mixing. The extractions were sedimented for 20 min at 30,000 x g at 4 °C. The supernatants were collected, frozen on dry ice, lyophilized to dryness, and subjected to LC/MS. $^{13}C9,^{15}N_3$-CTP was added for LC/MS analysis and peak areas of NTP and dNTPs were normalized to the peak area of $^{13}C9,^{15}N3$–CTP (*Chen et al., 2009*).

## Cloning and purification of *Ng* RNR subunits

The gene for the α subunit of *Ng* Ia RNR was optimized for expression in *Ec* and cloned into a vector with an N-terminal pET(His)$_6$SUMO-Kan-N. The protein was expressed, purified to homogeneity by Ni-affinity chromatography, and the tag cleanly removed using the SUMO protease (*Parker et al., 2018*). Subunit β was cloned and expressed with an N-terminal hexahistidine (His)$_6$- tag that resulted in homogeneous protein with variable amounts of metallo-cofactor. The diferric tyrosyl radical cofactor was self-assembled with $Fe^{2+}$ and $O_2$ to give 0.7–0.9 Y•s/β2 using the protocol of *Yee et al., 2003* except that 10 molar equivalents of $O_2$ instead of 3.5 was used and the ferrous ammonium sulfate was dissolved in buffer instead of water.

## Activity assay of *Ng* RNR

To optimize RNR activity a 1:1 ratio of $α_2$ and $β_2$ subunits was examined over a physiological concentration range of 0.01–10 µM. The specific activity (SA) of RNR was determined in a reaction mix of 100 µM *Ec* TR, 1 µM *Ec* TRR, and 0.2 mM NADPH at 37 °C. The SA for 0.01–0.12 µM subunits was measured with 1 mM GDP, 0.25 mM TTP, and 0.2 mM NADPH using the continuous spectrophotometric assay. The SA for 0.2–10 µM was measured with 1 mM 5-[$^3$H]-CDP, 3 mM ATP and 2 mM NADPH by the discontinuous radioactive assay. Assay detection methods followed the standard protocols (*Ge et al., 2003*; *Ravichandran et al., 2020*).

The stock solution of 0.48 mM PTC-672 was prepared in 50% DMSO aqueous solution and stored at –20 °C. The working solution was a 50-fold dilution of the stock solution to a final concentration of 9.6 µM in assay buffer with 1% DMSO. The concentration was determined by A348 nm with an $\varepsilon$ = 24.4 mM$^{-1}$cm$^{-1}$.

The assay for inhibition of *Ng* RNR by PTC-847 and PTC-672 was performed spectrophotometrically using the GDP/TTP protocol with the addition of various concentrations of PTC-847 (0.1–8 µM) or PTC-647 (0.025–2.5 µM). The inhibition of RNR by dATP was performed spectrophotometrically with the addition of various concentrations of dATP (2 nM to 1 µM). At concentrations of dATP from 50 to 200 µM, [$^3$H]-CDP/ATP and the discontinuous assay was used.

## Protocol for negative stain EM

Negative stain EM samples contained 1:1.5 α:β, 15 ng/µL protein. Concentrations of α and β were determined using A280 nm ($\varepsilon$ = 89,050 M$^{-1}$cm$^{-1}$ for α and 61,310 M$^{-1}$cm$^{-1}$ for β). αs (WT and mutants) and β were first combined on ice at 150 ng/µL and incubated for one min in the presence of specified nucleotides, also at 10 x concentration. The samples were then diluted tenfold in RNR assay buffer (50 mM HEPES pH 7.6, 15 mM $MgCl_2$, 1 mM EDTA) and 5 µL of the mixture was applied to carbon-coated 300 mesh Cu grids (Electron Microscopy Sciences) after glow discharging for 1 min at –15 mA in an EasiGlo glow discharger (Ted Pella). The protein was adsorbed to the grid for 1 min and excess liquid was removed by blotting. The grid was stained 3 x with 5 µL of 2% uranyl acetate (VWR) with blotting immediately after application for the first two rounds of staining. The final uranyl acetate stain was allowed to sit on the grid for 45 s before blotting. All blotting was completed manually with filter paper (Whatman, grade 40).

Screening images for the *Ng* RNR negative stain reconstruction were taken on a Tecnai Spirit (FEI) instrument with a XR16 camera (AMT) operated at 120 kV at ×68,000 magnification. Screening images for the H25R and S41L α variant samples were taken on a Tecnai F30 (FEI) operated at 200 kV equipped with an FEI Falcon II direct electron detector. Five to 10 grid squares were examined, and a representative field is shown for each protein sample. For the EM reconstruction of *Ng* RNR, images were taken on a Tecnai F20 Twin (FEI) operated at 120 kV equipped with a US4000 CCD detector (Gatan) at 1.79 Å/pix.

All processing for negative stain data sets was completed in Relion 3.0.8 (*Scheres, 2012*). Micrographs were imported and CTF-corrected prior to manual picking of 1000 particles subdivided into five classes to train Relion's autopicking algorithm. Autopicking yielded ~27,000 particles for the inactivated complex. These particles were 2D classified (100 classes, 25 iterations) and an initial 3D model was created (no symmetry imposed) prior to 3D classification (one class, 100 iterations, D2 symmetry) and 3D auto-refinement. Auto re-finement was completed on 12,469 particles and yielded a structure at 21 Å resolution (0.143 FSC). Following autorefinement, the inactive *Ec* RNR structure (PDB 5CNS) was docked into the *Ng* RNR EM map using the Chimera MaptoModel functionality (*Pettersen et al., 2004*; *Zimanyi et al., 2016*).

## Mass photometry

Mass photometry (interferometric scattering mass spectrometry) was performed on a Refeyn instrument using AcquireMP v2.4.1 and DiscoverMP v2.4.2. All movies were taken for a length of 60 s using the default parameters and Gaussian curves were fit to each histogram distribution using the DiscoverMP software. All samples were at a final concentration of 2.5 ng/µL protein with a 1:1 α:β ratio. Samples were incubated for 2 min at a concentration of 50 ng/µL at 37 °C prior to final dilution on the instrument and data recording. Concentrations of additives were as follows: 1 mM GTP, 0.25 mM TTP, 0.2 mM dATP, 3 mM ATP, 25 µM PTC-672 (0.45% DMSO), 8 µM PTC-847 (0.04% DMSO).

## In vivo studies using a mouse model of vaginal *Ng* infection

All animal studies were performed under IACUC approved protocols at AAALAC-certified animal facilities. Female ovariectomized Balb/c mice (Charles River, Wilmington, MA) were obtained at 3 weeks of age and acclimatized for ~1 week prior to initiation of a fitness or efficacy study. Day 0 was defined as the day mice were inoculated vaginally with ~1 × 10^9 bacteria suspended in 20 µL saline. A single 17-β-estradiol pellet (0.5 mg, 21day release) was then implanted in mice with a trochar on Day –2. Mice were then treated with three antibiotics to control commensal flora induced by the estradiol. Vancomycin HCl (0.6 mg) and streptomycin sulfate (0.3 mg) were administered intraperitoneally and trimethoprim sulfate (0.8 mg) was administered by oral gavage twice daily from Day –2 to Day 1 of study. On Day 2 of the study the administration of all antibiotics except streptomycin sulfate was stopped. Streptomycin sulfate was then administered daily from Day 2 throughout the remainder of the study.

## Fitness testing of PTC-672-resistant *Ng* in the mouse model of vaginal *Ng* infection

The relative fitness cost associated with PTC-672 resistance was determined in vivo by competition experiments using the validated mouse model of vaginal *Ng* infection described above. This study evaluated the relative fitness of the PTC-672 resistant strains Iso 13477 Strep^R-PTC-672^H25R and Iso 13479 Strep^R-PTC-672^H25R in competition with the isogenic PTC-672 susceptible parent strains Iso 13477 Strep^R and Iso 13479 Strep^R. On Day 0 of the study, mice pre-treated as described above were inoculated vaginally with either the Strep^R parent strain, the isogenic Strep^R-PTC672^H25R strain, or a mixture of the Strep^R parent and the isogenic Strep^R-PTC672^H25R strains suspended in saline and swabbed daily over a one week period. At each sampling time point, the numbers of total (susceptible+ resistant bacteria) and resistant bacteria were determined by plating onto non-selective and PTC-672-supplemented plates, respectively. Relative fitness was expressed as the CI, calculated as the ratio of bacterial burdens between the PTC-672 resistant and susceptible strains at each time point divided by the baseline ratio at the beginning of the experiment.

## In vivo efficacy studies using the mouse model of vaginal *Ng* infection

The efficacy of an increasing single oral dose of PTC-672 administered by oral gavage was evaluated in the mouse model described above. The antibiotics, ciprofloxacin and ceftriaxone, were used as

positive controls for efficacy. Efficacy was defined as sustained clearance of infection for 5 days post dosing in 10/10 mice in a treatment group.

On Day 0 of the study, mice pre-treated as described above were inoculated vaginally with Iso 13477 Strep[R] or Iso 13479 Strep[R] bacteria suspended in saline. On Day 2 mice were randomized into treatment groups based on Day 1 CFU counts. All mice were treated on Day 2 with a single oral dose of either vehicle comprising 0.5% hydroxypropyl methylcellulose and 0.1% Tween 80, ciprofloxacin, ceftriaxone, or PTC-672. In the Iso 13477 Strep[R] portion of the study, PTC-672 was tested at doses of 10, 15, 20, 25, and 30 mg/kg and ciprofloxacin at 30 mg/kg. In the Iso 13479 Strep[R] portion of the study, PTC-672 was tested at doses of 30 and 60 mg/kg and ciprofloxacin at 100 mg/kg. Mice were vaginally swabbed daily using a sterile swab over a one-week period. At each sampling time point, the numbers of resistant bacteria were determined by plating onto PTC-672-supplemented plates and the level of infection was determined for each test mouse in each treatment group.

## Acknowledgements

This project had been funded in part by the Wellcome Trust through a Seeding Drug Discovery award (097753) and by the National Institute of Health Grants R35 GM126982 (CLD) and NIH Pre-Doctoral Training grant T32 GM007287 (TSL) and NSF GRFP 2017246757 (TSL). CLD is a Howard Hughes Medical Institute Investigator. Chang Cui was supported by NIH grant GM47274 to Daniel G Nocera, Department of Chemistry at Harvard University and NIH grant GM29595 (JS). The authors wish to thank Richard Sheridan for assembling the data and getting the manuscript in submittable format.

## Additional information

### Competing interests

Jana Narasimhan, Suzanne Letinski, Stephen P Jung, Aleksey Gerasyuto, Jiashi Wang, Michael Arnold, Guangming Chen, Jean Hedrick, Melissa Dumble, Gary Karp, Arthur Branstrom: was employed by PTC Therapeutics when the work was performed and received salary and compensation during their tenure. The other authors declare that no competing interests exist.

### Funding

| Funder | Grant reference number | Author |
| --- | --- | --- |
| Wellcome Trust | 097753 | Arthur Branstrom |
| National Institutes of Health | GM126982 | Catherine L Drennan |
| National Institutes of Health | GM007287 | Talya Levitz |
| National Science Foundation | 2017246757 | Talya Levitz |
| National Institutes of Health | GM29595 | JoAnne Stubbe |
| Howard Hughes Medical Institute | | Catherine L Drennan |
| Harvard University | | Chang Cui |
| National Institutes of Health | GM047274 in Nocera lab | Chang Cui |

The funders had no role in study design, data collection and interpretation, or the decision to submit the work for publication.

### Author contributions

Jana Narasimhan, JoAnne Stubbe, Gary Karp, Conceptualization, Data curation, Formal analysis, Investigation, Methodology, Project administration, Resources, Software, Supervision, Validation,

Visualization, Writing – original draft, Writing – review and editing; Suzanne Letinski, Stephen P Jung, Data curation, Formal analysis, Investigation, Methodology, Software, Validation, Visualization; Aleksey Gerasyuto, Jiashi Wang, Michael Arnold, Guangming Chen, Melissa Dumble, Catherine L Drennan, Conceptualization, Data curation, Formal analysis, Investigation, Methodology, Project administration, Resources, Software, Supervision, Validation, Visualization, Writing – review and editing; Jean Hedrick, Conceptualization, Data curation, Formal analysis, Investigation, Methodology, Resources, Software, Validation, Visualization; Kanchana Ravichandran, Conceptualization, Data curation, Formal analysis, Investigation, Methodology, Software, Validation, Visualization; Talya Levitz, Conceptualization, Data curation, Formal analysis, Investigation, Methodology, Software, Validation, Visualization, Writing – original draft, Writing – review and editing; Chang Cui, Conceptualization, Data curation, Formal analysis, Investigation, Methodology, Software, Validation, Visualization, Writing – review and editing; Arthur Branstrom, Conceptualization, Data curation, Formal analysis, Funding acquisition, Investigation, Methodology, Project administration, Resources, Software, Supervision, Validation, Visualization, Writing – original draft, Writing – review and editing

### Author ORCIDs
Jana Narasimhan ⬤ http://orcid.org/0000-0001-5621-8592
Stephen P Jung ⬤ http://orcid.org/0000-0002-9276-175X
Catherine L Drennan ⬤ http://orcid.org/0000-0001-5486-2755
JoAnne Stubbe ⬤ http://orcid.org/0000-0001-8076-4489
Arthur Branstrom ⬤ http://orcid.org/0000-0002-5917-6372

### Ethics
Animal studies were done according to procedures reviewed and approved by the Rutgers Institutional Animal Care and Use Committee (IACUC). The IACUC protocol ID used was I12-075-12.

### Decision letter and Author response
Decision letter https://doi.org/10.7554/eLife.67447.sa1
Author response https://doi.org/10.7554/eLife.67447.sa2

## Additional files

### Supplementary files
• Supplementary file 1. PTC-847 and PTC-672 MIC values across 206 *N. gonorrhoeae* strains collected at the Public Health England. Two isolates gave elevated PTC-847 or PTC-672 MIC values. However, these isolates were sensitive to all other antibiotics. Due to the selectivity of the PTC compounds, these two strains will be genotyped to confirm they are *Neisseria* species. PTC-compound susceptibility testing was performed in accordance with the Clinical and Laboratory Standards Institute (CLSI) M07-A9 guideline (*Clinical and Laboratory Standards Institute, 2012*).

• Supplementary file 2. Time -dependent kill of *Ng* upon PTC-847 (A) or PTC-672 treatment (B). At PTC-847 or PTC-672 concentrations of ≥1 x MIC, a > 3 log reduction in measured *Ng* 13477 colony forming units per milliliter of culture (CFU/mL) was observed. The time-dependent kill assays were performed in accordance with the CLSI M26-A guideline (*Clinical and Laboratory Standards Institute, 1999*).

• Supplementary file 3. Time-dependent post antibiotic effect observed with PTC-847. *Ng* 13477 was pre-treated with PTC-847 at 4 X MIC for 2 h then allowed to grow in culture media supplemented with PTC-847 at 0, 0.25, 0.5, or 1 X MIC (Data for 0, 0.25 x MIC, 0.5 x MIC, and 1 x MIC are indicated in black, grey, purple, and blue respectively). A > 6 log reduction in measured *Ng* 13477 CFU/mL was observed at 20 h post pre-treatment in the absence of PTC-847. The post antibiotic effect assay was performed in accordance with the CLSI M26-A guideline (*Clinical and Laboratory Standards Institute, 1999*).

• Supplementary file 4. PTC-847 inhibition of DNA synthesis in *N. gonorrhoeae* strain 13477. We followed the incorporation of the radiolabeled precursors into total nucleic acids (black circles), DNA (red squares), and protein (blue triangles). In *Ng*, only radiolabeled uracil is incorporated into DNA or RNA, which was also shown to be the case for *Nm* (*Jyssum and Jyssum, 1979*).

• Supplementary file 5. Frequency of resistance to PTC-847. By plating at high cell density on agar plates containing PTC-847 at 4, 8, 16 and 32-fold MIC, Ng 13477 exhibited a spontaneous or acquired frequency of resistance to PTC-847 on the order of $10^{-8}$. The colonies obtained were

passaged multiple times on PTC-847-containing plates to obtain a stable PTC-847-resistant strain (PTC-847$^R$).

• Supplementary file 6. PTC-847$^R$ strain exhibits no cross resistance to existing antibiotics. Susceptibility of the WT *Ng* 13477 strain and the PTC-847$^R$ strain were measured for a wide variety of antibiotics having different modes of action. MICs were determined in accordance with the CLSI M07-A9 guideline (*Clinical and Laboratory Standards Institute, 2012*). The PTC-847$^R$ strain was equally sensitive to all classes of antibiotics as the susceptible WT *Ng* 13477 strain, except for resistance to PTC-847. The PTC-847 MIC for WT *Ng* 13477 was 0.05 µg/mL compared to 15.6 µg/mL for the PTC-847$^R$ strain.

• Supplementary file 7. Nucleotide pools measured in WT *Ng* 13477. The susceptible WT Ng 13477 strain was grown to log phase and treated with DMSO or PTC-847 at 1 X MIC for 1 h. Nucleotides were extracted in acidified acetonitrile/$H_2O$ (65:35) followed by centrifugation. The supernatant was lyophilized and subjected to LC/MS. $^{13}C9,^{15}N3$–CTP was added for LC/MS analysis. Peak areas of NTP and dNTPs were normalized to the peak area of $^{13}C9,^{15}N3$–CTP.

• Supplementary file 8. Characterization of purified *Ng* RNR. (A) Activity assay of *Ng* RNR. To optimize RNR activity, a 1:1 ratio of $\alpha_2$ and $\beta_2$ subunits was examined over a physiological concentration range of 0.01–10 µM and its specific activity (SA) determined in a reaction mix of 100 µM *Ec* thioredoxin (TR), 1 µM *Ec* thioredoxin reductase (TRR), and 0.2 mM NADPH at 37 °C. The SA for 0.01–0.12 µM subunits was measured with 1 mM GDP, 0.25 mM TTP, and 0.2 mM NADPH using a spectrophotometric assay whereas the 0.2–10 µM SA was measured with 1 mM 5-[$^3$H]-CDP, 3 mM ATP and 2 mM NADPH by radioactive assay. Fitting the data from 0.01 to 0.12 µM using $v = \frac{V_{max}[Subunits]}{[Subunits]+K_m}$ gave $V_{max}$ = 1300 nmol min$^{-1}$ mg$^{-1}$ and $K_m$ = (0.030 ± 0.006) µM. (n = 2 replicates for each concentration). (B) EPR spectra of the tyrosyl radical. The X-band EPR spectrum of the tyrosyl radical of *Ec* $\beta_2$ (red) and *Ng* $\beta_2$ (black) at 77 K. Microwave power: 35 dB. Modulation: 1.5 Gauss. Modulation frequency: 100 kHz. Time constant: 20.48ms.

• Supplementary file 9. dATP inhibition of *Ng* RNR. The continuous spectrophotometric assay with GDP/TTP (Figure S4) was used for concentrations of dATP to 25 µM and NADPH concentration was 0.2 mM. The discontinuous radioactive assay with [5-$^3$H] CDP (1 mM) and ATP (3 mM) was used for dATP at 50–200 µM and NADPH concentration at 2 mM (see ref 38). The assay with mutant αs (H25R and S41L) used the spectrophotometric assay with GDP/TTP.

• Supplementary file 10. PTC-672 and PTC-847 do not inhibit a representative panel of normally occurring intestinal organisms. The panel was constructed to test for inhibition of normally occurring intestinal organisms (*Thursby and Juge, 2017*). Susceptibility testing was performed in accordance with the Clinical and Laboratory Standards Institute (CLSI) M07-A9 guideline (*Clinical and Laboratory Standards Institute, 2012*). PTC-672 was tested against a more extensive panel of organisms.

• Supplementary file 11. Commensal *Neisseria* are less sensitive to PTC-672 than *Ng* 13477. Two commensal *Neisseria* 'type strains' swabbed from the oropharynx of healthy volunteers ($^†$*Berger, 1971* and ‡*Riou and Guibourdenche, 1987*) were tested for susceptibility to inhibition in accordance with the Clinical and Laboratory Standards Institute (CLSI) M07-A9 guideline (*Clinical and Laboratory Standards Institute, 2012*).

• Supplementary file 12. The distribution of the identified RNR types in various bacteria (*Lundin et al., 2009*) that were used to assess antibacterial activity of the PTC compounds.

• Transparent reporting form

### Data availability

All data generated or analyzed during this study are included in the manuscript and supplemental section.

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
