## [Editor Report]

This paper is of interest to biochemists and those focused on development of novel antibiotics. The authors present two small molecules that specifically target the essential ribonucleotide reductase of the causative agent of gonorrhea. Biochemical, biophysical, and biological data support the efficacy of these molecules both in vitro and in mouse models. Overall, this is a comprehensive study providing insights that may guide the development of new therapies for gonorrhea.

---

## [Decision Letter]

**Decision letter after peer review:**

Thank you for submitting your article "Ribonucleotide reductase, a novel target for gonorrhea" for consideration by *eLife*. Your article has been reviewed by 2 peer reviewers, and the evaluation has been overseen by a Reviewing Editor and Bavesh Kana as the Senior Editor. The reviewers have opted to remain anonymous.

Essential revisions:

1) The authors should provide additional structural or biochemical evidence to support the conclusion that the inhibitor induces inhibited RNR tetramer formation and/or to identify the inhibitor binding site.

2) The claims about pathogen selectivity and evasion of drug resistance must also be revised or more appropriately justified by additional experimental data.

*Reviewer #1 (Recommendations for the authors):*

Specific suggestions for improvement:

The introduction is very detailed and could be better tailored to a general audience.

In general, the microbiology findings and the biochemical/structural work could be more fully integrated. For example, is there a molecular-level explanation for why a mechanism of action involving potentiation of inhibited tetramers would be selective for the Ng pathogen – and nontoxic to humans?

It would also be nice to see a comparison to existing drugs or other drug candidates for MIC and resistance frequency values. Additionally, more explanation of methods for microbiology assays is needed. A reference is provided for a standard protocol, but I think it is important to state exactly how the experiments were performed (# replicates, error analysis, etc) to allow a reader to reproduce the experimental work.

The basis for concluding that resistance is less likely to occur with these inhibitors than with other drugs is not clear. In general, how can one draw this conclusion about drug resistance in humans based on work done in bacterial culture or in animal models?

The conclusion that commensal bacteria are not affected by the inhibitor is not sufficiently supported. It seems that only four selected anaerobic gut microbiome constituents were tested – and these likely have different RNRs than Ng. It would be ideal to look at vaginal microbiome organisms or other aerobic bacteria known to rely on class I RNR enzymes.

Finally, information about exactly how the drug interacts with Ng RNR is lacking. A structure of the enzyme bound to the inhibitor would both fill in missing gaps about how the drug works and enable expansion to other systems or elaboration to more effectively target Ng and minimize resistance.

---

## [Author Response]

Essential revisions:1) The authors should provide additional structural or biochemical evidence to support the conclusion that the inhibitor induces inhibited RNR tetramer formation and/or to identify the inhibitor binding site.

In this paper, we are putting forward the proposal that these compounds inhibit RNR through stabilization of the inactive tetrameric form of the enzyme. We agree that we do not have the data needed to draw a conclusion about the mechanism of inhibition at this time. We apologize if our phrasing suggested otherwise. We have rewritten the corresponding sections to make it clear what we know and what we do not know. We have also provided additional data that further probe the mechanism of action of these compounds. Importantly, we present data showing that the compounds work synergistically with dATP, enhancing the inhibitory effects of dATP. These new data support the proposal that the formation of the dATP-inhibited tetramer ring state is a part of the mechanism of inhibition. However, we also show new data that RNR variants that are unable to form the tetrameric ring state are still inhibited by these compounds. Collectively these data indicate that multiple mechanisms of inhibition may be play at least in vitro. Sorting out the details of these multiple mechanism will take further studies. For this paper, we have added new data figures S7-S9 and rewritten the results and discussion.

2) The claims about pathogen selectivity and evasion of drug resistance must also be revised or more appropriately justified by additional experimental data.

We have included supplemental Table S5 highlighting a representative panel of commensal organisms that are not inhibited by the novel PTC analogs. Table S2 highlights the frequency of resistance for the compounds using *Neisseria gonorrhoeae* compared to ciprofloxacin and ceftriaxone. Examining the experiments for competitive infection, both the in vitro data (Figure 8) and the in vivo data (Figure 9) demonstrate that the *Ng* PTC-672^H25R^ strains are less fit to compete with isolates having wildtype RNRs.

Reviewer #1 (Recommendations for the authors):Specific suggestions for improvement:The introduction is very detailed and could be better tailored to a general audience.

The introduction has some minor edits to help with the understanding of the various complex topics covered in this manuscript. One of the challenges noted is that the paper covers several aspects of drug discovery, including chemical structure SAR, mechanism of action elucidation, biochemical analysis of the *Ng* RNR, antibacterial testing, and animal efficacy. This is a true multi-disciplinary effort and brings novel information to the scientific community.

In general, the microbiology findings and the biochemical/structural work could be more fully integrated. For example, is there a molecular-level explanation for why a mechanism of action involving potentiation of inhibited tetramers would be selective for the Ng pathogen – and nontoxic to humans?

This question and statement are not quite accurate. The authors don’t claim that potentiation of the inactive tetrameric state of RNR is selective for *Ng* pathogens. It is possible that other bacterial class Ia RNRs could be inhibited through tetrameric state potentiation as well. However, many bacteria have multiple classes of RNRs such that they are not dependent on their class Ia RNR for survival. Neisseria has only one RNR, a class Ia enzyme, which could explain its sensitivity to these compounds. Antibacterial MICs data indicate that organisms with multiple RNRs have elevated MICs (Supplemental Tables S5 and S7). In terms of why these compounds do not potentiate formation of inactive tetramers in the case of human RNR. We showed

It would also be nice to see a comparison to existing drugs or other drug candidates for MIC and resistance frequency values.

We agree. Text was included for frequency of resistance, and this information was summarized in Table S2. Table S3 depicts the MIC values for a complete panel of antibiotics against the wild-type and PTC-847^R^ isolates. No cross-resistance was observed with any of the antibiotics tested.

Additionally, more explanation of methods for microbiology assays is needed. A reference is provided for a standard protocol, but I think it is important to state exactly how the experiments were performed (# replicates, error analysis, etc) to allow a reader to reproduce the experimental work.

Details have been added to more thoroughly indicate how the microbiology assays were performed. We believe those outside the field of study should now be able to perform the assays.

The basis for concluding that resistance is less likely to occur with these inhibitors than with other drugs is not clear. In general, how can one draw this conclusion about drug resistance in humans based on work done in bacterial culture or in animal models?

Historically, frequency of resistance and bactericidal determinations are good predictors for drug success in humans. Clinical trials are the only definitive way to validate success. However, antibacterial drug discovery is one of the few fields where the antibacterial activity in animal models directly correlates with human data (along with corresponding DMPK data). Our preclinical data suggest that an isolate that becomes resistant to the PTC analogs has less probability to become multi-drug resistant because those variants are at a competitive disadvantage relative to organisms lacking the RNR mutation. In drug development, it is known that any antibiotic that makes it into the clinic will have organisms that become resistant. The question is not “if”, but “when” resistance will occur. The frequency of resistance numbers are favorable, as are the data for fitness.

The conclusion that commensal bacteria are not affected by the inhibitor is not sufficiently supported. It seems that only four selected anaerobic gut microbiome constituents were tested – and these likely have different RNRs than Ng. It would be ideal to look at vaginal microbiome organisms or other aerobic bacteria known to rely on class I RNR enzymes.

Data have been included in Table S5 showing the MICs against commensals compared to solithromycin. In this table, you will note that ALL the isolates were inhibited by solithromycin, and NONE were inhibited by PTC-672 except the *Neisseria* species. Table S7 indicates the known RNRs in various bacteria listed by genus. These data are only correlative suggesting drug sensitivity is associated with the presence of only RNR Ia. Further experimentation will be necessary to elucidate the exact resistance mechanism in other species.

Finally, information about exactly how the drug interacts with Ng RNR is lacking. A structure of the enzyme bound to the inhibitor would both fill in missing gaps about how the drug works and enable expansion to other systems or elaboration to more effectively target Ng and minimize resistance.

Structural studies are in progress. They are nontrivial and will be the subject of a future manuscript if we are successful.